# ZERO GENERALIZATION ERROR THEOREM FOR RANDOM INTERPOLATORS VIA ALGEBRAIC GEOMETRY

## ABSTRACT

We theoretically demonstrate that the generalization error of interpolators for machine learning models under teacher-student settings becomes 0 once the number of training samples exceeds a certain threshold. Understanding the high generalization ability of large-scale models such as deep neural networks (DNNs) remains one of the central open problems in machine learning theory. While recent theoretical studies have attributed this phenomenon to the implicit bias of stochastic gradient descent (SGD) toward well-generalizing solutions, empirical evidences indicate that it primarily stems from properties of the model itself. Specifically, even randomly sampled interpolators—parameters that achieve zero training error—have been observed to generalize effectively. In this study, under a teacher–student framework, we prove that the generalization error of randomly sampled interpolators becomes exactly zero once the number of training samples exceeds a threshold determined by the geometric structure of the interpolator set in parameter space. As a proof technique, we leverage tools from algebraic geometry to mathematically characterize this geometric structure.

## 1 INTRODUCTION

Triggered by the success of deep neural networks, increasing attention has been paid to methods that employ a large model and yet achieve excellent generalization performance while perfectly fitting the training data (Simonyan & Zisserman, 2015; Zhang et al., 2017). Such learning models that attain exact fit to the training set are referred to as *interpolators*. Explaining the performance of these interpolators constitutes one of the central challenges in contemporary deep learning theory, and several distinct lines of work have emerged to address this phenomenon (Neyshabur et al., 2015; Bartlett et al., 2017; Golowich et al., 2018).

Among these lines of works trying to explain the generalization performance of interpolators, one of the leading explanations has been the *implicit bias* induced by learning algorithms, typically stochastic gradient descent (SGD) and its variants. This perspective posits that SGD guides parameters toward generalizing solutions in parameter space, such as the minimum $L^2$-norm solution (Yun et al., 2021). Although this line of work has achieved notable progress, several challenges and limitations have been identified. First, both theoretical and empirical studies have reported cases where implicit bias does not necessarily enhance generalization (Dauber et al., 2020; Vyas et al., 2024; Farhang et al., 2022). Second, most of the existing theoretical analyses are restricted to simplified settings, such as linear models or two-layer neural networks, due to a fundamental difficulty in identifying the solution for non-convex optimization problems (Gunasekar et al., 2017; Soudry et al., 2018; Arora et al., 2019; Lyu & Li, 2020; Chizat & Bach, 2020; Vardi, 2023; Cattaneo et al., 2024).

In parallel, a growing body of recent work has emphasized that *model-based properties*, independent of the algorithm's implicit bias, can enhance the generalization performance of interpolators. In particular, it is becoming increasingly evident that SGD behaves like a uniform sampling on a set of interpolators. Valle-Pérez et al. (2019) empirically showed that SGD behaves similarly to uniform sampling from the set of interpolators, which implies that the models generalize well due to a natural bias of interpolators. This finding was further supported by Mingard et al. (2021) who corroborated the similarity between SGD and uniform sampling over interpolators. Additionally, Chiang et al. (2023) examined the generalization properties of both SGD-optimized parameters and randomly

sampled interpolators, demonstrating that the generalization ability of DNNs is largely independent of the employed optimization algorithm.

An important challenge within this line of research on interpolators is to provide a theoretical foundation for the empirical observation above. Specifically, we pose the following question:

*Can the strong generalization performance of interpolators be explained*

*by a model-based theory that does not rely on the implicit bias of algorithms?*

## 1.1 Our Result

In this study, we develop a model-based theory and mathematically prove that an interpolator can achieve zero-generalization error even with a limited amount of training data. Specifically, under the teacher–student learning framework and a random interpolator, we demonstrate that the minimal number of samples required to attain zero generalization error, what we term the *strong sample complexity*, is finite and admits an explicit upper bound. In short, we have the following informal statement:

**Theorem 1** (Informal statement of Theorem 2)**.** *The following holds with probability* $1$*:*

*(Strong sample complexity)*

$\leq$ *(Dimension of parameter space)* $-$ *(Dimension of true parameter set)* $+ 2$.

This result implies that even when the model becomes large, the strong sample complexity can remain small provided that the dimension of the true parameter set is also large. This yields a purely mathematical theory demonstrating that an interpolator can achieve sufficiently strong generalization ability without reliance on any specific optimization algorithm.

Our analysis develops this theory by applying the concept of *real analytic sets*, originating in algebraic geometry, to study the relationship between the geometric structure of the interpolator set and that of the true parameter set. Specifically, real analytic sets describe the intersections of zero sets of analytic functions in real space. This framework is crucial for determining the dimension of the interpolator set, since interpolators are precisely characterized as the zeros of the loss function evaluated on the training data.

We summarize our contributions as follow:

1. We develop a model-based theory for interpolators, thereby providing a theoretical justification for the empirical finding that generalization error can be explained solely by the structure of the model.
2. Specifically, we theoretically demonstrate the strong sample complexity required for an interpolator to achieve zero generalization error (Theorem 2). This phenomenon of attaining zero generalization error under finite data is a discovery unique to interpolators.
3. Through several concrete examples, we uncover new insights, including cases where over-parameterization does not affect sample complexity, and even instances where it reduces sample complexity (Theorem 5, 6).
4. We introduce the new concept of real analytic sets from algebraic geometry into machine learning theory, establishing a novel theoretical foundation.

## 1.2 Related Works

**Generalization ability of interpolators.** There are several works studying the generalization ability of interpolators. Buzaglo et al. (2024) showed that in the teacher-student setting, the sample complexity of quantized DNN does not explicitly depend on the number of parameters when the parameter is randomly sampled from its interpolator set, using PAC-Bayes like analysis. While they study the standard sample complexity, the number of data necessary for the generalization error become less than $\epsilon$, we study the number of data by which the generalization error becomes exactly zero. Valle-Pérez et al. (2019) studied generalization error when the parameter is sampled from a uniform distribution on the interpolator set. They showed that the PAC-Bayes generalization bound guarantees the good generalization of such a predictor. Theisen et al. (2021) proved that a large proportion of interpolators in two layer ReLU neural network have good generalization ability

in binary classification tasks. Yang et al. (2021) studied the uniform generalization bound on the interpolator of random feature models. They showed that as the number of features increases, the generalization error decreases. Belkin (2021) investigated the generalization ability of interpolators in kernel methods in over-parameterization. He showed that such interpolators exhibit an implicit bias toward simple functions, ensuring that the generalization error does not increase with over-parameterization.

**Geometric structure of interpolator set.** Finally, we list several studies investigating the geometric landscape of the set of interpolators. Cooper (2021) clarified the relation between the number of training data and the dimension of the set of the interpolators. We use the similar analysis for deriving Theorem 2. We remark that while he studied the cases in which a noise exists in the data-generating process, our study is about noiseless cases. Fukumizu et al. (2019) studied the landscape of interpolator set by investigating three ways for a wider student network producing the same output as the teacher network, which is also similar to our analysis for deriving Theorem 5 and 6.

## 1.3 Notation

For $n \in \mathbb{N}$, $[n]$ denotes $\{1, 2, ..., n\}$. We denote the set of non-negative real numbers as $\mathbb{R}_{\geq 0}$ and non-negative integers as $\mathbb{Z}_{\geq 0}$. We denote the Euclidean norm as $\|\cdot\|$, and the $\ell^1$-norm as $\|\cdot\|_1$. For a set $S \subset \mathbb{R}^d$, the distance between the set $S$ and a point $\omega \in \mathbb{R}^d$ is denoted $\|\omega - S\| = \inf\{\|\omega - s\| \mid s \in S\}$.

## 2 Preliminaries

### 2.1 Regression Problem with Teacher-Student Setting

We formalize our problem setup, a regression problem with a teacher-student setting.

**Data generating process.** We define the input space $\mathcal{X} \subset \mathbb{R}^m$ as an $m$-dimensional real analytic manifold (defined later in Section 3.2.1) and the output space $\mathcal{Y}$. The corresponding output $y \in \mathcal{Y}$ for an input $x \in \mathcal{X}$ is generated by the *teacher model*, a function $f^*(\cdot; \theta^*) : \mathcal{X} \to \mathcal{Y}$, as

$$y = f^*(x; \theta^*), \tag{1}$$

where $\theta^* \in \mathbb{R}^{d^*}$ is a fixed $d^*$-dimensional parameter. Suppose that we observe $n$ samples $\{(x_i, y_i)\}_{i=1}^n$, where $(x_i, y_i) \in \mathcal{X} \times \mathcal{Y}$ and each $x_i$ is drawn independently and identically according to a probability measure $\mathcal{D}$ on $\mathcal{X}$, and $y_i$ follows the teacher model (1) with given $x_i$. We note that this noiseless setting is common in theoretical works under the teacher-student framework, as in Tian (2017); Safran & Shamir (2018); Xu & Du (2023).

**Regression problem.** Using the samples $\{(x_i, y_i)\}_{i=1}^n$, we consider training a model $f : \mathcal{X} \times \Theta \to \mathcal{Y}$ called *student model*

$$f(x; \theta), \ x \in \mathcal{X}, \theta \in \Theta,$$

where $\theta$ is an $\mathbb{R}^{d_\Theta}$-valued parameter to be trained and $\Theta \subset \mathbb{R}^{d_\Theta}$ is a compact $d_\Theta$-dimensional real analytic manifold. We focus on the regression problem and consider the squared loss function $\ell(y, y') = \frac{1}{2}\|y - y'\|^2$. Note that this setup can be extended to a general analytic loss function. We define a training error $L_n(\theta) := \frac{1}{n}\sum_{i=1}^n \ell(y_i, f(x_i; \theta))$ and a generalization error $L(\theta) := \mathbb{E}_{x \sim \mathcal{D}}[\ell(y, f(x; \theta))]$.

### 2.2 Real Analytic Function

We introduce an important concept for our analysis, a real analytic function.

**Definition 1** (Real analytic function). A real analytic function is a function $f : U \to \mathbb{R}$, where $U$ is an open subset of $\mathbb{R}^d$, such that for every point $\theta^{(0)} \in U$, the function $f$ can be expressed as a convergent power series which converges in a neighborhood of $\theta^{(0)}$:

$$f(\theta) = \sum_{j=0}^{\infty} \sum_{\alpha: \|\alpha\|_1 = j} c_\alpha (\theta - \theta^{(0)})^\alpha,$$

where $\theta = (\theta_1, \cdots, \theta_d) \in U$ and $\alpha = (\alpha_1, \cdots, \alpha_d) \in \mathbb{Z}_{\geq 0}^d$ is a multi-index of non-negative integers by which we define $\theta^\alpha = \theta_1^{\alpha_1} \cdots \theta_d^{\alpha_d}$. When the output of $f$ is a vector, $f$ is called a real analytic function if each of its components is a real analytic function.

Throughout our analysis, we assume that the student model is a real analytic function with respect to the parameter $\theta$ and the input data $x$.

**Assumption 1** (Real Analytic Student Model). *We assume that $f(x; \theta)$ is a real analytic function with respect to $\theta \in \Theta$ and $x \in \mathcal{X}$.*

A wide range of machine learning models satisfy this assumption: for example, a fully connected deep neural network or attention mechanism whose activation function is analytic, such as sigmoid, softmax, hyperbolic tangent, and so on.

## 3    Interpolator and Teacher-Equivalent Set

We introduce key concepts that form the foundation of our analysis. The first is the predictor based on a *random interpolator*, with particular emphasis on implementations using neural networks. The second is a *teacher-equivalent set*, a notion that plays a central role in our generalization theory.

### 3.1    Predictor with Random Interpolator

We consider a predictor that interpolates the training data by sampling a parameter from a distribution supported on the set of parameters that perfectly interpolates the training data.

In preparation, we consider an *interpolating parameter set* (IPS), satisfying zero training error, *i.e.*,

$$\widehat{\Theta}_n := \{\theta \in \Theta \mid L_n(\theta) = 0\} = \{\theta \in \Theta \mid \ell(y_i, f(x_i; \theta)) = 0, \forall i \in [n]\}.$$

Note that $\widehat{\Theta}_n$ is a random set from the random training sample of $\{(x_i, y_i)\}_{i=1}^n$.

Next, we consider randomly sampling a parameter $\theta$ from $\widehat{\Theta}_n$, following a distribution $\mathbb{P}(\theta \mid \widehat{\Theta}_n)$ that is absolutely continuous with respect to the uniform distribution on $\widehat{\Theta}_n$. We then consider a predictor with the sampled parameter:

$$f(x; \widehat{\theta}_n), \ \ \widehat{\theta}_n \sim \mathbb{P}(\cdot \mid \widehat{\Theta}_n).$$

This random predictor interpolates the training samples with probability 1, i.e., $\mathbb{P}(L_n(\widehat{\theta}_n) = 0) = 1$.

### 3.2    Teacher-Equivalent Set (TES)

We define the notion of a teacher-equivalent set (TES). We say that a parameter $\theta \in \Theta$ of a student model is *teacher-equivalent* when $f(x; \theta) = f^*(x; \theta^*)$ holds for every $x \in \mathcal{X}$. Then, the *teacher-equivalent set* (TES) is a set of the teacher-equivalent parameters:

$$\bar{\Theta} := \{\theta \in \Theta \mid f(x; \theta) = f^*(x; \theta^*), \forall x \in \mathcal{X}\}.$$

Intuitively, TES is the true parameter set, i.e., the set of parameters of the student model that replicate the teacher model. In contrast to the IPS $\widehat{\Theta}_n$, the TES $\bar{\Theta}$ is a non-random set.

In our analysis, we assume that the TES is not empty as a realizability assumption.

**Assumption 2** (Realizability). *The TES $\bar{\Theta}$ is non-empty, i.e., there exists a parameter $\theta^\circ \in \Theta$ satisfying $f^*(x; \theta^*) = f(x; \theta^\circ)$ for every $x \in \mathcal{X}$.*

This realizability assumption is natural for the teacher-student setup with smaller teachers (Tian, 2017; Safran & Shamir, 2018; Xu & Du, 2023). In fact, this assumption is satisfied in the case of fully-connected deep neural networks, as is shown in Section 5.

The TES $\bar{\Theta}$ can be considered to be the population version of the IPS $\widehat{\Theta}_n$. Moreover, as the sample size $n$ diverges to infinity, a student model from $\widehat{\Theta}_n$ interpolates all possible data points in the sample space $\mathcal{X} \times \mathcal{Y}$, causing $\widehat{\Theta}_n$ to asymptotically converge to $\bar{\Theta}$. Therefore, as $n$ increases, $\widehat{\Theta}_n$ monotonically shrinks and approaches $\bar{\Theta}$. Indeed, we show that $\widehat{\Theta}_n \setminus \bar{\Theta}$ is a null set in the meaning of Lebesgue measure on $\widehat{\Theta}_n$ in Section 4. Figure 1 illustrates this intuition.

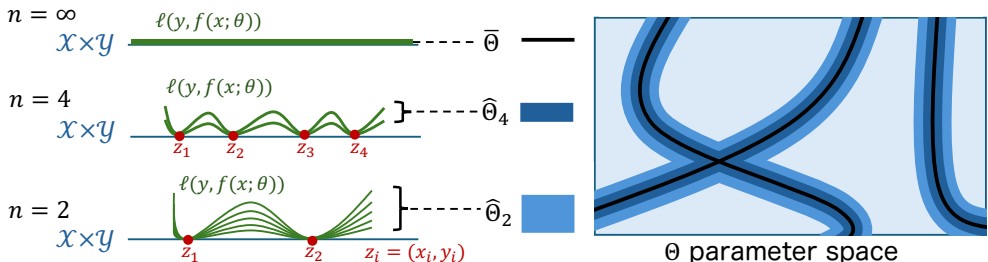

Figure 1: Illustration of the IPS $\widehat{\Theta}_n$ approaches the TES $\bar{\Theta}$. As $n$ increases, fewer parameters $\theta$ achieve the interpolator, i.e. $\ell(y_i, f(x_i; \theta)) = 0, \forall i = 1, ..., n$, causing $\widehat{\Theta}_n$ to converge to $\bar{\Theta}$. Note that while $\widehat{\Theta}_n$ and $\bar{\Theta}$ are plotted on the same plane in the right panel, $\widehat{\Theta}_n$ is actually higher dimensional than $\bar{\Theta}$.

### 3.2.1 Dimension of TES

We define a dimension of the TES $\bar{\Theta}$. First, we define an analytic manifold and its dimension.

**Definition 2** (Real Analytic manifold). A $d'$-dimensional real analytic manifold is a topological manifold $\Omega$ ($\subset \mathbb{R}^d$) equipped with a $d'$-dimensional local coordinate system $\{(U_i, \varphi_i)\}_i$ where $U_i$ is an open subset of $\mathbb{R}^d$ and $\varphi_i : U_i \rightarrow U_i'$ for an open subset $U_i'$ of $\mathbb{R}^{d'}$ such that if $U_i \cap U_j \neq \varnothing$ holds, the transition maps $\varphi_j \circ \varphi_i^{-1} : \varphi_i(U_i \cap U_j) \rightarrow \varphi_j(U_i \cap U_j)$ are real analytic functions.

We remark that in the Euclidean space $\mathbb{R}^d$, all the open subsets are real analytic manifolds in the meaning of a standard topology and their dimensions correspond to $d$.

We now state the definition of the dimension of the TES.

**Definition 3** (Dimension of TES). The dimension of the TES $\bar{\Theta}$, denoted as $d_{\bar{\Theta}}$, is defined as the maximum dimension of a real analytic manifold $\Omega$ contained in $\bar{\Theta}$.

Intuitively, the dimension of the TES is a number of unrestricted parameters. For example, in a model where a unique parameter $\theta' \in \Theta$ achieves teacher equivalence, such as a linear regression model, the dimension of the TES is 0 since the TES is a singleton.

## 4 Generalization Error Analysis

We analyze the generalization error of a predictor with a random interpolator defined in Section 3.

### 4.1 Main Theorem

We investigate the number of samples required for a predictor with a random interpolator to achieve zero generalization error. To facilitate the analysis, we define the important notion, the *strong sample complexity*, which quantifies the necessary sample size.

**Definition 4** (Strong sample complexity for a random interpolator). For the sampled interpolator $\widehat{\theta}_n \sim \mathbb{P}(\cdot \mid \widehat{\Theta}_n)$, its strong sample complexity for the generalization error $L(\widehat{\theta}_n)$ is defined as

$$k(\widehat{\Theta}_n) := \min \left\{ n \in \mathbb{N} \mid \mathbb{P}(L(\widehat{\theta}_n) = 0 \mid \widehat{\Theta}_n) = 1 \right\}.$$

The strong sample complexity is the minimum number of data necessary for $\widehat{\theta}_n$ to achieve zero generalization error completely; in other words, by which almost all interpolators from $\widehat{\Theta}_n$ are teacher equivalent. This notion is a stronger version of the ordinary sample complexity, which is the sample size necessary to achieve the generalization error smaller than some positive value.

Before our main theorem, we put an assumption with regard to the distribution $\mathcal{D}$ of the input data $x$.

**Assumption 3** (Data distribution). *The probability measure $\mathcal{D}$ of an input $x$ is absolutely continuous with respect to that of a uniform distribution on $\mathcal{X}$.*

We now show our main result on the generalization error of a predictor with a random interpolator. We provide its proof in Appendix B.

**Theorem 2** (Strong sample complexity: general case). *Suppose that Assumptions 1, 2, and 3 hold. Then, the strong sample complexity for a random interpolator satisfies the following with probability 1 in terms of $\mathcal{D}$:*

$$k(\widehat{\Theta}_n) \leq d_\Theta - d_{\bar{\Theta}} + 1.$$

This theorem states that with no less than $d_\Theta - d_{\bar{\Theta}} + 1$ training samples, a predictor with a random interpolator achieves zero generalization error with probability 1. This fact illustrates the view that the strong sample complexity is determined by the dimension of the TES. We note that $d_{\bar{\Theta}}$ becomes large when using practical machine learning models such as deep neural networks, although there exist simple cases in which $\bar{\Theta}$ is a singleton and therefore $d_{\bar{\Theta}} = 0$ as discussed below. We will present applications of Theorem 2 to deep neural networks, where $d_{\bar{\Theta}}$ becomes large and the resulting bound is non-vacuous, in Section 5.

As a simple case, we consider a model in which a unique parameter $\theta' \in \Theta$ achieves teacher equivalence. A typical example is a linear regression model where the covariance matrix of inputs is non-degenerate. In this case, $d_{\bar{\Theta}}$ is a singleton set, meaning its dimension is zero and the upper bound of strong sample complexity is $d_\Theta + 1$. This result is provided in the following corollary without proof.

**Corollary 3.** *Consider the setup of Theorem 2. Further, suppose that $\bar{\Theta}$ is a singleton set. Then, the strong sample complexity of $\widehat{\Theta}_n$ satisfies the following with probability 1 in terms of $\mathcal{D}$:*

$$k(\widehat{\Theta}_n) \leq d_\Theta + 1.$$

### 4.2 Analysis for Near Interpolator Case

For more practical scenarios, we consider the situation where a parameter is not an exact interpolator but a near interpolator. For $\varepsilon > 0$, the $\varepsilon$-neighborhood of $\widehat{\Theta}_n$ is defined as $\widehat{\Theta}_{n,\varepsilon} := \{\theta \in \Theta \mid \|\theta - \widehat{\Theta}_n\| \leq \varepsilon\}$. Then, we define a *predictor with a random near interpolator* as follows:

$$f(x; \widehat{\theta}_{n,\varepsilon}), \ \ \widehat{\theta}_{n,\varepsilon} \sim \mathbb{P}(\cdot \mid \widehat{\Theta}_{n,\varepsilon}),$$

where $\mathbb{P}(\cdot \mid \widehat{\Theta}_{n,\varepsilon})$ the uniform distribution on $\widehat{\Theta}_{n,\varepsilon}$.

As a strong sample complexity of a predictor sampled according to $\mathbb{P}(\cdot \mid \widehat{\Theta}_{n,\varepsilon})$, we have the following result immediately. We postpone its proof to Appendix C.

**Proposition 4** (The generalization error of the near interpolator). *Fix small $\varepsilon > 0$. Suppose that Assumption 1, 2, and 3 hold. Moreover, we assume that there exists a universal constant $q > 0$ such that for every $x \in \mathcal{X}$, $f(x; \theta)$ is $q$-Lipschitz continuous in $\theta$. If the number of data satisfies $n \geq d_\Theta - d_{\bar{\Theta}} + 1$, then the following holds with probability at least $1 - O(\varepsilon)$ with respect to $\mathcal{D}$ and $\mathbb{P}(\cdot \mid \widehat{\Theta}_{n,\varepsilon})$:*

$$L(\widehat{\theta}_{n,\varepsilon}) \leq (q\varepsilon)^2.$$

The assumption of Lipschitz continuity is general in theoretical research in machine learning. It is satisfied in most of the analytic functions including deep neural networks or convolutional neural networks whose activation function is a sigmoidal function or transformers with a softmax function.

### 4.3 Proof Outline of Theorem 2 with Real Analytic Sets

We prove Theorem 2 by showing that $\widehat{\Theta}_n \setminus \bar{\Theta}$ becomes a null set with respect to the Lebesgue measure on $\widehat{\Theta}_n$ when $n \geq d_\Theta - d_{\bar{\Theta}} + 1$. This is established by showing that the geometrical dimension of $\widehat{\Theta}_n \setminus \bar{\Theta}$ becomes less than that of $\bar{\Theta}$, using tools from the theory of real analytic sets. Once this fact

is established, it follows that $\ell(y, f(x; \widehat{\theta}_n)) = 0$ for every $x \in \mathcal{X}$ and for almost every $\widehat{\theta}_n \in \widehat{\Theta}_n$, which directly implies $\mathbb{E}[\ell(y, f(x; \widehat{\theta}_n))] = 0$.

In preparation, we introduce the key notion, a real analytic set. The IPS $\widehat{\Theta}_n$ and TES $\bar{\Theta}$ are obviously real analytic sets.

**Definition 5** (Real analytic set). A set of zeros of real analytic functions is called a *real analytic set*; that is, for real analytic functions $f_1, \cdots f_n : \Theta \to \mathbb{R}$, a real analytic set is defined as

$$\{\theta \in \Theta \mid f_1(\theta) = 0, \cdots, f_n(\theta) = 0\}.$$

We further define the dimension of a real analytic set, which is the most important concept in our analysis. Intuitively, the dimension of a real analytic set is a number of free parameters for making the function be zero. We note that from the definition, the dimension of a real analytic set is an integer.

**Definition 6** (Dimension of real analytic set). The dimension of a real analytic set $A$ is defined as the maximum $d'$ such that $A$ contains a real analytic manifold of dimension $d'$.

An important property of real analytic sets is that they can be treated locally in much the same way as algebraic varieties, i.e., sets defined as the common zeros of polynomials. This stems from the fact that an analytic function can be locally expressed as a convergent power series, that is, as a sum of polynomials. Moreover, analogous results from complex algebraic geometry have been established via the technique of complexification, developed by Whitney & Bruhat (1959). As a consequence, many results concerning algebraic varieties also hold for real analytic sets.

Among the most fundamental results concerning the dimensions of algebraic varieties is Krull's principal ideal theorem (Hartshorne, 1977). Informally, Krull's principal ideal theorem states that the dimension of the solution set of $n$ polynomial equations decreases by $n$. For example, let $\theta = (\theta_1, \theta_2, \theta_3) \in \mathbb{R}^3$ and consider the solution set of two equations $\theta_1 + \theta_2 = 0$ and $\theta_2 \theta_3 = 0$. The resulting set is $\{(a, -a, 0) \mid a \in \mathbb{R}\} \cup \{(0, 0, a) \mid a \in \mathbb{R}\}$, which consists of two straight lines in $\mathbb{R}^3$, and hence has dimension 1.

By applying arguments analogous to Krull's principal ideal theorem to real analytic sets, the dimension of $\widehat{\Theta}_n \setminus \bar{\Theta}$ becomes $d_\Theta - n$, since $\widehat{\Theta}_n$ is defined as a solution set of $n$ equations. Moreover, because $\widehat{\Theta}_n \supset \bar{\Theta}$ and the dimension of $\bar{\Theta}$ is $d_{\bar{\Theta}}$, it follows that the dimension of $\widehat{\Theta}_n \setminus \bar{\Theta}$ is strictly smaller than that of $\bar{\Theta}$ whenever $n \geq d_\Theta - d_{\bar{\Theta}} + 1$. Consequently, $\widehat{\Theta}_n \setminus \bar{\Theta}$ becomes a null set with respect to the Lebesgue measure on $\widehat{\Theta}_n$, which establishes the theorem.

## 5 Application for practical models

We study the generalization error of the predictor with a random interpolator for specific models by applying Theorem 2. Specifically, we analyze the dimension $d_{\bar{\Theta}}$ of TES $\bar{\Theta}$ for each model and derive the upper bound of strong sample complexity.

### 5.1 Deep Linear Neural Network

We first study deep linear neural networks (DLNNs), whose activation is an identical mapping.

**Teacher-Student Setup.** The student model is an $L$-layer DLNN with parameters $w^{(\ell)} \in \mathcal{W}^{(\ell)}$ and $b^{(\ell)} \in \mathcal{B}^{(\ell)}$, where $\mathcal{W}^{(\ell)} \subset \mathbb{R}^{m_\ell \times m_{\ell-1}}$ and $\mathcal{B}^{(\ell)} \subset \mathbb{R}^{m_\ell}$ are sufficiently large compact sets for $\ell = 1, ..., L$ as

$$f(x; \theta) = w^{(L)}(w^{(L-1)} \cdots (w^{(1)}x + b^{(1)}) \cdots + b^{(L-1)}) + b^{(L)}.$$

We denote $\theta = \{w^{(\ell)}, b^{(\ell)}\}_{\ell=1}^L$. We remark that $m_0$ is the input dimension and $m_L$ is the output dimension of the network.

We consider that the teacher model is also a DLNN with $L^*$-layers and $m_\ell^*$ width for $\ell$-th layer. Specifically, $f^*$ is a DLNN with parameters $\theta^* = \{w^{(\ell)*}, b^{(\ell)*}\}_{\ell=1}^{L^*}$, where $w^{(\ell)*} \in \mathbb{R}^{m_\ell^* \times m_{\ell-1}^*}$ and $b^{(\ell)*} \in \mathbb{R}^{m_\ell^*}$. Suppose that the teacher model is smaller than the student model, that is, we have $L \geq L^*$, $m_\ell \geq m_\ell^*$ ($1 \leq \ell \leq L^* - 1$), and $m_\ell \geq m_{L^*-1}^*$ ($L^* \leq \ell \leq L - 1$).

**Results.** We study the strong sample complexity of the interpolator for DLNNs. This result is derived by constructing a subset of $\bar{\Theta}$ and calculating its dimension, as is proved in Appendix D.

**Theorem 5.** *Let the dimension of the parameter of the teacher DLNN be $d^*$. Suppose that Assumption 3 holds. Then the strong sample complexity of DLNN satisfies the following with probability 1 in terms of $\mathcal{D}$:*

$$k(\widehat{\Theta}_n) \le d^* + 1.$$

This theorem shows that the strong sample complexity of the student network remains bounded by a constant, regardless of its size. Thus, to achieve good generalization, one may employ an arbitrarily large model without concern for its capacity, consistent with common practice in applied settings.

## 5.2 Fully Connected Deep Neural Network

We study the learning problem with general fully connected deep neural networks (FCDNNs), whose activation is a general analytic function.

**Teacher-Student Setup.** The student model we train is an $L$-layer FCDNN defined with parameters $w^{(\ell)} \in \mathcal{W}^{(\ell)}$ and $b^{(\ell)} \in \mathcal{B}^{(\ell)}$, where $\mathcal{W}^{(\ell)} \subset \mathbb{R}^{m_\ell \times m_{\ell-1}}$ and $\mathcal{B}^{(\ell)} \subset \mathbb{R}^{m_\ell}$ are sufficiently large compact sets for $\ell = 1, ..., L$ as

$$f(x;\theta) = w^{(L)}\sigma(w^{(L-1)} \cdots \sigma(w^{(1)}x + b^{(1)}) \cdots + b^{(L-1)}) + b^{(L)}.$$

We denote $\theta = \{w^{(\ell)}, b^{(\ell)}\}_{\ell=1}^{L}$. Here, $\sigma$ denotes the analytic activation function.

Similar to the previous section, we consider the teacher model as a FCDNN with parameters $\theta = \{w^{(\ell)*}, b^{(\ell)*}\}_{\ell=1}^{L^*}$ for $w^{(\ell)*} \in \mathbb{R}^{m_\ell^* \times m_{\ell-1}^*}$ and $b^{(\ell)*} \in \mathbb{R}^{m_\ell^*}$ and with the same activation function $\sigma$ as the student FCDNN. We assume that the width of student FCDNN is larger than or equal to that of the teacher FCDNN, that is, we have $L = L^*$ and $m_\ell \ge m_\ell^*$ ($1 \le \ell \le L^* - 1$).

**Result.** We now present our result on the strong sample complexity of the interpolator for FCDNNs. In the same way as Theorem 5, this result is obtained by constructing a subset of $\bar{\Theta}$ and computing its dimension, with the detailed proof provided in Appendix E.

**Theorem 6.** *Suppose that Assumption 3 holds. Then the strong sample complexity of FCDNN satisfies the following with probability 1 in terms of $\mathcal{D}$:*

$$k(\widehat{\Theta}_n) \le \sum_{\ell=1}^{L} m_\ell^*(m_{\ell-1} + 1) + 1.$$

## 6 Experiments

### 6.1 Near Interpolators for FCDNN+Teacher-Student Setup

We experimentally investigate properties of near interpolators introduced in Section 4.2, under the same conditions as in Section 5.2, that is, both the student and teacher models are fully-connected deep neural networks (FCDNNs) for evaluating the strong sample complexity. We sample random near interpolators by *Guess and Check* (G&C) algorithm (Chiang et al. (2023), details are described in Appendix F.1) until the training loss falls below 0.01 for 1000 times, yielding 1000 random near interpolator samples. We employ G&C as the sampling algorithm for near interpolators, rather than stochastic gradient descent (SGD), since SGD may introduce a bias toward specific subregions of the interpolator set $\widehat{\Theta}_n$ and thus fail to satisfy absolute continuity of $\mathbb{P}(\cdot \mid \widehat{\Theta}_n)$ with respect to the uniform distribution. Additional details are provided in Appendix F.2.

**Result.** Figure 2 reports the test losses of random interpolators for each network. Across all the three models, the theoretical upper bound of the strong sample complexity in Theorem 6 is consistent with the experimental results, suggesting that it provides a sufficient condition for the generalization of random interpolators. We remark that test losses does not go to exactly zero since we only sample the *near* interpolators (see Proposition 4).

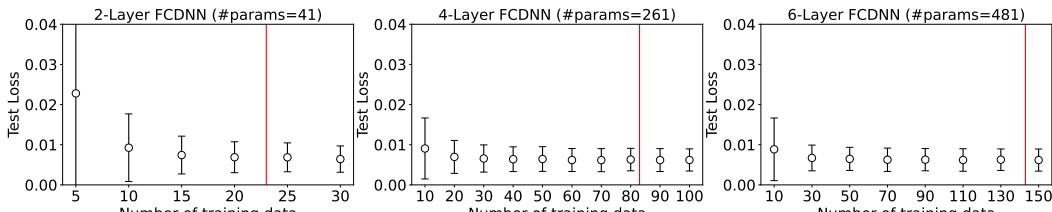

Figure 2: Test losses of random near interpolators on 2-layer FCDNN (left), 4-layer FCDNN (middle), and 6-layer FCDNN (right). The vertical axis represents the test loss, while the horizontal axis corresponds to the number of training data. The error bars indicate the standard deviation over 1000 trials for each training sample size. The red vertical line is the theoretical upper bound of the strong sample complexity in Theorem 6.

## 6.2 Near Interpolators for Large Models+MNIST

We study properties of near interpolators in a more practical setting than in Section 6.1, that is, we utilize LeNet (Lecun et al., 1998) and MNIST dataset (LeCun et al., 2010). We sample random near interpolators using the Xavier's initialization (Glorot & Bengio, 2010) and the Adam optimizer with a batch size of 1024, running until the training loss on the full batch of the selected MNIST subset falls below 0.01. This procedure is repeated 2000 times, yielding 2000 random near interpolator samples. As noted in the previous section, the bias of SGD should in principle be avoided by employing the G&C algorithm; however, due to computational constraints, we approximate it by sampling with Adam. Nevertheless, the experimental results obtained using the *pattern search* algorithm, which is an alternative procedure for generating random near interpolators without implicit bias, analogous to the G&C algorithm, presented in Appendix F.4, indicate that the findings remain largely consistent regardless of the choice of algorithm.

To estimate the dimension $d_{\bar{\Theta}}$ of $\bar{\Theta}$, we approximately sample from $\bar{\Theta}$ by training LeNet on the full MNIST dataset using Adam with a batch size of 1024, until the full-batch training loss falls below 0.01. This process is repeated 30000 times, yielding 30000 approximate samples from $\bar{\Theta}$. Then, we estimated the dimension of the manifold on which this 30000 samples lie by using the scikit-dimension package (Bac et al., 2021). More details are described in Appendix F.3.

**Result.** Figure 3 reports the test losses of random near interpolators for LeNet on MNIST. The estimated upper bound, indicated by the red line, is consistent with the experimental results, since it provides a sufficient number of data for the generalization of random near interpolators. We remark that the test losses do not converge exactly to zero, since we only sample *near* interpolators (see Proposition 4).

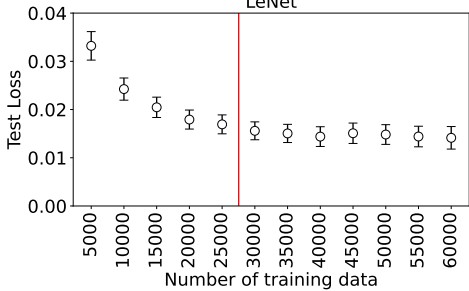

Figure 3: Test losses of random near interpolators on LeNet. The vertical axis represents the test loss, while the horizontal axis corresponds to the number of training data. The error bars indicate the standard deviation over 2000 trials for each training sample size. The red vertical line is the estimated upper bound of the strong sample complexity $d_{\Theta} - d_{\bar{\Theta}} + 1$.

## 7 Conclusion

This study investigates the generalization error of randomly sampled interpolators in general machine learning models. Within the framework of a teacher–student regression problem, we show that the generalization error of a randomly sampled interpolator becomes exactly zero once the number of training samples exceeds a threshold determined by the dimension of the teacher-equivalence set (TES). Moreover, we establish that for both deep linear neural networks and fully connected deep neural networks, the strong sample complexity does not explicitly depend on the size of the network.

## LLM usage

We utilized large language models (LLMs) for a translation aid to ensure natural and fluent academic writing and to assist in debugging and resolving errors in our experimental code.

## Impact Statement

This paper presents work whose goal is to advance the field of Machine Learning. There are many potential societal consequences of our work, none which we feel must be specifically highlighted here.

## Ethics Statement

This work is purely theoretical and does not involve experiments with human subjects, personal data, or other sensitive information. Our aim is to contribute to the advancement of machine learning by providing theoretical guidance that may help inform the design of more efficient model architectures. We hope that these insights will have a positive impact on the future development of machine learning technology.

## Reproducibility Statement

All experimental settings are described in detail in Section 6 and Appendix F. To further support reproducibility, we provide the full source code used to run these experiments as a supplementary material.

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

# A    Basic Concepts of Real Analytic Sets

We introduce basic concepts and results in real analytic sets. Consistently with the main contents, let $\Theta \subset \mathbb{R}^{d_\Theta}$ be a compact real analytic manifold.

## A.1    Irreducible sets

First, we see the property of irreducible sets. We define the irreducibility of a real analytic set as follows.

**Definition 7** (Irreducibility). A real analytic set $X$ is called *irreducible* if it is not the union of two strictly smaller real analytic sets.

We show two important theorems about the irreducible sets.

**Theorem 7** (Proposition 11, Whitney & Bruhat (1959)). *For a real analytic set $X$, there uniquely exists a locally finite family $\{S_\lambda\}_{\lambda \in \Lambda}$ of irreducible real analytic subsets of $X$ such that $X = \bigcup_\lambda S_\lambda$ and $S_\lambda \not\subset S_\mu$ for $\lambda, \mu \in \Lambda$ with $\lambda \neq \mu$.*

**Theorem 8** (Corollary, Section 8, Whitney & Bruhat (1959)). *An irreducible real analytic set contains no proper real analytic subset of its same dimension.*

From Theorem 7, we can show the following immediately.

**Proposition 9.** *A real analytic set $X$ defined on a compact space $\Theta$ can be decomposed into a finite number of irreducible real analytic sets.*

*Proof.* From Theorem 8, there exists a locally finite family of irreducible real analytic sets $\{S_\lambda\}_\lambda$ such that $X = \bigcup_\lambda S_\lambda$ holds. Since $X$ is a closed subset of $\Theta$, $X$ is also compact. Therefore, since $\{S_\lambda\}_\lambda$ is locally finite cover and $X$ is compact,, we can choose a finite number of subsets from $\{S_\lambda\}_\lambda$ covering $X$. These subsets are the desired irreducible real analytic sets. $\square$

## A.2    Regularity

Second, we introduce the notion of regularity. We consider a real analytic set $X \subset \Theta$ and define $d := \dim(X)$.

**Definition 8** (Analytic isomorphism). Two sets $X_1, X_2 \in \mathbb{R}^d$ are said to be analytically isomorphic if there exists an analytic isomorphic function from $X_1$ to $X_2$.

We define the regularity of a point in a real analytic set as follows. This definition follows Guaraldo et al. (1986).

**Definition 9** (Regularity of a point). For a fixed point $x \in X$, $x$ is said to be regular if there exists an open neighborhood $U$ of $x$ such that $U$ is analytically isomorphic to some open set $V$ in $\mathbb{R}^d$.

In other words, the neighborhood of $x$ can be locally regarded as an open set in $\mathbb{R}^d$.

As an important property of regular points, we show the following theorem.

**Theorem 10** (Theorem 1, Chapter III, Narasimhan (1966)). *The set of regular points in $X$ is dense in $X$.*

# B    Proof of Theorem 2

*Proof.* First, we show that, regardless of the choice of $x_1, ..., x_n$, the dimension of $\widehat{\Theta}_{n+1} \setminus \bar{\Theta}$ becomes less than that of $\widehat{\Theta}_n \setminus \bar{\Theta}$ by probability 1, in terms of the probabilistic choice of $x_{n+1}$. Second, we show that the dimension of $\widehat{\Theta}_n \setminus \bar{\Theta}$ becomes at least $n - 1$ less than that of $\widehat{\Theta}_1 \setminus \bar{\Theta}$ with probability 1, in terms of the probabilistic choice of $x_1, ..., x_n$. Finally, we complete the proof by characterizing the properties of $\widehat{\theta}_n$ sampled from $\widehat{\Theta}_n$.

**Step0: The dimension of $\widehat{\Theta}_1$ becomes one less than that of $\Theta$.** Before proceeding to the main part of the proof, we first establish a straightforward preliminary result: the dimension of $\widehat{\Theta}_1$ is $d_\Theta - 1$. This follows from a standard result of analytic sets, like Lojaciewicz's Structure Theorem (Krantz & Parks, 1992), which states that an analytic set defined on $\mathbb{R}^{d_\Theta}$ can be decomposed into $d_\Theta - 1, ...,$ and 1-dimensional manifolds.

**Step1: The dimension of $\widehat{\Theta}_{n+1}$ becomes less than that of $\widehat{\Theta}_n$ with probability 1.** We prove this by using the knowledge of analytic varieties as we see in section A. As we show in Proposition 9, $\widehat{\Theta}_n$ ($n \geq 1$) can be decomposed into a finite number of irreducible real analytic sets. We define a sequence of irreducible real analytic sets

$$T_{n,1}, ..., T_{n,k} \subset \widehat{\Theta}_n$$

such that they are not included in $\bar{\Theta}$. If it does not hold, $\bar{\Theta} = \widehat{\Theta}_n$ holds, and the proof is completed, so we assume their existence. Choose one arbitrarily from the sequence of sets and denote it by $T$. Let the dimension of $T$ be $d_T$. $d_T$ may vary depends on the choice of $T_{n,i}$, but it does not affect the following proof.

We prepare some useful notion of sets. We denote

$$\mathcal{X}_T := \{x \in \mathcal{X} | \{\theta \in \widehat{\Theta}_n \mid \ell(f^*(x;\theta^*), f(x;\theta)) = 0\} \supset T\}.$$

Furthermore, we fix a regular point $\theta' \in T$ and define

$$\mathcal{X}_{\theta'} := \{x \in \mathcal{X} \mid \ell(f^*(x;\theta^*), f(x;\theta')) = 0\}.$$

Since $\ell(f^*(x;\theta^*), f(x;\theta')) = 0$ holds for any $x \in \mathcal{X}_T$, we have $\mathcal{X}_T \subset \mathcal{X}_{\theta'}$. In addition, $\mathcal{X}_{\theta'}$ is a real analytic set with respect to $x$ since $\mathcal{X}$ is a subset of $\mathbb{R}^m$ and $\mathcal{X}_{\theta'}$ is a set of zeros of real analytic functions with respect to $x$.

We show that the dimension of $\mathcal{X}_T$ is less than $m$, which is the dimension of $\mathcal{X}$, and its Lebesgue measure on $\mathcal{X}$ is equal to 0. To this aim, we assume that the dimension of $\mathcal{X}_{\theta'}$ is $m$ and prove that it is a conflict. As we see in Theorem 10, the set of regular points in $T$ is dense. Hence, we can choose one of the regular points of $T$ and its open neighborhood $U \subset T$. From the regularity, $U$ can be regarded as an open set $\widehat{U}$ in $\mathbb{R}^{d_T}$. In the same way, we can choose a regular point in $\mathcal{X}_{\theta'}$, and its neighborhood $V$ can be identified with an open set $\widehat{\mathcal{X}}_{\theta'}$ in $\mathbb{R}^m$. Hence, we have

$$\ell(f^*(x;\theta^*), f(x;\theta)) = 0, \ \forall x \in \widehat{\mathcal{X}}_{\theta'}, \ \forall \theta \in \widehat{U}.$$

So, the identity theorem shows that $\ell(y, f(x;\theta))$ is equal to 0 for every $x \in \mathbb{R}^m$ and $\theta \in \mathbb{R}^{d_T}$. However, it means that $T$ is included in $\bar{\Theta}$, so it is a conflict to the assumption to the definition of $T$. Hence, the dimension of $\mathcal{X}_{\theta'}$ is less than $m$. Since $\mathcal{X}_T \subset \mathcal{X}_{\theta'}$ holds, the dimension of $\mathcal{X}_T$ is also less than $m$. So the Lebesgue measure of $\mathcal{X}_T$ on $\mathcal{X}$ is 0.

We define an independent variable $x_{n+1}$ generated from the distribution $\mathcal{D}$. Since $\mathcal{D}$ is absolutely continuous to a uniform distribution on $\mathcal{X}$, we have from the discussions above,

$$\{\theta \in \widehat{\Theta}_n \mid \ell(f^*(x_{n+1};\theta^*), f(x_{n+1};\theta)) = 0\} \cap T_{n,i} \subsetneq T_{n,i}$$

with probability 1 for every $i = 1, ..., k$. Since $\{\theta \in \widehat{\Theta}_n \mid \ell(f^*(x_{n+1};\theta^*), f(x_{n+1};\theta)) = 0\}$ is a real analytic set and $T_{n,i}$ is irreducible, the dimension of $\{\theta \in \widehat{\Theta}_n \mid \ell(f^*(x_{n+1};\theta^*), f(x_{n+1};\theta)) = 0\}$ is less than that of $T_{n,i}$ from Theorem 8. Since this holds for all the components $T_{n,i}$ of $\widehat{\Theta}_n$ and

$$\widehat{\Theta}_{n+1} \setminus \bar{\Theta} = (\widehat{\Theta}_n \cap \{\theta \in \widehat{\Theta}_n \mid \ell(f^*(x_{n+1};\theta^*), f(x_{n+1};\theta)) = 0\}) \setminus \bar{\Theta}$$

$$= \bigcup_{i=1}^{k} (\{\theta \in \widehat{\Theta}_n \mid \ell(f^*(x_{n+1};\theta^*), f(x_{n+1};\theta)) = 0\} \cap T_{n,i}) \setminus \bar{\Theta}, \text{ and}$$

$$\widehat{\Theta}_n \setminus \bar{\Theta} = \bigcup_{i=1}^{k} T_{n,i} \setminus \bar{\Theta}$$

hold, the dimension of $\widehat{\Theta}_{n+1} \setminus \bar{\Theta}$ is less than that of $\widehat{\Theta}_n \setminus \bar{\Theta}$ with probability 1.

**Step2: The dimension of $\widehat{\Theta}_n$ becomes at least $n-1$ less than that of $\widehat{\Theta}_1$ with probability 1.** We denote by $\widehat{\Theta}(x_1, \ldots, x_n)$ the zero set determined by $x_1, \ldots, x_n$, and define the event that the dimension of $\widehat{\Theta}(x_1, \ldots, x_n, x_{n+1})$ is less than that of $\widehat{\Theta}(x_1, \ldots, x_n)$ as $A(x_1, ..., x_{n+1})$. What we aim to show is that, when $x_1, \ldots, x_n$ are i.i.d. generated random variables from $\mathcal{D}$,

$$\mathbb{P}(A(x_1, ..., x_{n+1}) \cap A(x_1, ..., x_n) \cap \cdots \cap A(x_1, x_2)) = 1.$$

From step1, the probability of a random variable $x_2$ causing an event $A(x_1', x_2)$ under given that $x_1 = x_1'$ for a fixed $x_1'$ is given by

$$\mathbb{P}(A(x_1', x_2) \mid x_1') = 1$$

for any $x_1' \in \mathcal{X}$. Therefore, we have for i.i.d. random variables $x_1, x_2 \sim \mathcal{D}$,

$$\mathbb{P}(A(x_1, x_2)) = \int_{x_1' \in \mathcal{X}} \mathbb{P}(A(x_1', x_2) \mid x_1') \cdot \mathrm{d}\mathbb{P}(x_1') = 1, \tag{2}$$

where $\mathbb{P}(A(x_1', x_2) \mid x_1')$ denotes the probability measure of the event where $A(x_1', x_2)$ occurs under given that $x_1 = x_1'$ and $\mathrm{d}\mathbb{P}(x_1')$ denotes the probability measure of the event where $x_1 = x_1'$ holds for a fixed $x_1' \in \mathcal{X}$.

Next, observe that for i.i.d. random variables $x_1, x_2, x_3 \sim \mathcal{D}$,

$$
\begin{aligned}
\mathbb{P}\left(\overline{A(x_1, x_2, x_3) \cap A(x_1, x_2)}\right) &= \mathbb{P}\left(\overline{A(x_1, x_2, x_3)} \cup \overline{A(x_1, x_2)}\right) \\
&\leq \mathbb{P}\left(\overline{A(x_1, x_2, x_3)}\right) + \mathbb{P}\left(\overline{A(x_1, x_2)}\right) \\
&= \mathbb{P}\left(\overline{A(x_1, x_2, x_3)}\right) \tag{3}
\end{aligned}
$$

holds from (2). Moreover, we have

$$\mathbb{P}\left(\overline{A(x_1, x_2, x_3)}\right) = \int_{x_1', x_2' \in \mathcal{X}} \mathbb{P}\left(\overline{A(x_1', x_2', x_3)} \mid x_1', x_2'\right) \cdot \mathrm{d}\mathbb{P}(x_1', x_2'),$$

where $\mathbb{P}\left(\overline{A(x_1', x_2', x_3)} \mid x_1', x_2'\right)$ denotes a probability measure of the event where $A(x_1', x_2', x_3)$ does not occur under given that $x_1 = x_1', x_2 = x_2'$ and $\mathrm{d}\mathbb{P}(x_1', x_2')$ denotes the probability measure of the event where $x_1 = x_1'$ and $x_2 = x_2'$ occur for fixed $x_1', x_2' \in \mathcal{X}$. Since $\mathbb{P}(A(x_1', x_2', x_3) \mid x_1', x_2') = 1$ holds for any $x_1', x_2' \in \mathcal{X}$ from step1, we have

$$\int_{x_1', x_2' \in \mathcal{X}} \mathbb{P}\left(\overline{A(x_1', x_2', x_3)} \mid x_1', x_2'\right) \cdot \mathrm{d}\mathbb{P}(x_1', x_2') = 0 \cdot 1 = 0$$

and therefore, we have from (3),

$$\mathbb{P}(A(x_1, x_2, x_3) \cap A(x_1, x_2)) = 1.$$

By continuing this argument inductively, we obtain

$$\mathbb{P}(A(x_1, ..., x_{n+1}) \cap A(x_1, ..., x_n) \cap \cdots \cap A(x_1, x_2)) = 1$$

as desired.

**Step3: The properties of $\widehat{\theta}_n$ sampled from $\widehat{\Theta}_n$.** From step0 and step2, when $n \geq d_\Theta - d_{\bar{\Theta}} + 1$, the dimension of $\widehat{\Theta}_n \setminus \bar{\Theta}$ is less than that of $\bar{\Theta}$ and therefore, less than that of $\widehat{\Theta}_n$. As a consequence, the Lebesgue measure of $\widehat{\Theta}_n \setminus \bar{\Theta}$ on $\widehat{\Theta}_n$ corresponds to 0. Since $\mathbb{P}(\cdot|\widehat{\Theta}_n)$ is absolutely continuous to the uniform distribution on $\widehat{\Theta}_n$, if we sample $\widehat{\theta}_n \sim \mathbb{P}(\cdot|\widehat{\Theta}_n)$, we have

$$\widehat{\theta}_n \in \bar{\Theta}$$

with probability 1, which completes the proof. □

# C  PROOF OF PROPOSITION 4

## C.1  THE ASYMPTOTIC FORM OF THE VOLUME OF A NEIGHBORHOOD

In the proof, we utilize a theory developed by Federer (1996) with regard to the asymptotic form of the volume of an $\varepsilon$-neighborhood of a manifold. For completeness of the paper, we present basic concepts of this theory in this section.

First, we present a notion of Minkowski content, which defines an asymptotic form of the volume of an $\varepsilon$-neighborhood for small $\varepsilon$.

**Definition 10** (Minkowski content)**.** Let $A$ be a Lebesgue measurable set in $\mathbb{R}^n$ and $\mathrm{Vol}(A)$ be its Lebesgue measure on $\mathbb{R}^n$. If there exists the following value for $S \subset \mathbb{R}^n$, we call it $m$-dimensional Minkowski content and denote it as $\mathcal{M}^m(S)$:

$$\lim_{\varepsilon \to 0+} \frac{\mathrm{Vol}(\{x \mid \|x - S\| \le \varepsilon\})}{\alpha(n - m)\varepsilon^{n-m}},$$

where $\alpha(m)$ is the Lebesgue measure of a unit sphere in $\mathbb{R}^m$.

**Definition 11** (Rectifiable)**.** Let $S$ be a subset of a metric space $X$ and $m$ be a positive integer. Then $S$ is called *$m$-rectifiable* if and only if there exists a Lipschitzian function mapping some bounded subset of $\mathbb{R}^m$ onto $S$.

From the definition, an $m$-dimensional smooth compact manifold in $\mathbb{R}^n$ is $m$-rectifiable.

**Definition 12** (Hausdorff measure)**.** Let $X$ be a metric space with distance $\rho$ and $S$ be a Carathéodory-measurable set in it. We consider a $\delta$-covering $\{U_i^\delta\}$ of $S$ as

$$S \supset \bigcup_{i=1}^{\infty} U_i^\delta, \mathrm{diam}(U_i^\delta) \le \delta,$$

where $\mathrm{diam}(U_i^\delta) := \sup_{x,y \in U_i^\delta} \rho(x, y)$. We define

$$\mathcal{H}_\delta^m(S) := \inf_{\{U_i^\delta\}} \sum_{i=1}^{\infty} \mathrm{diam}(U_i^\delta)^m.$$

Then we call the following value *$m$-dimensional Hausdorff measure* of $S$.

$$\mathcal{H}^m(S) := \lim_{\delta \to 0} \mathcal{H}_\delta^m(S).$$

We note that while the Hausdorff measure is not necessarily a finite value, when $S$ is an $m$-dimensional smooth compact manifold in $\mathbb{R}^n$, the $m$-dimensional Hausdorff measure of it becomes a positive finite value (see section 3.2.46 in Federer (1996) for instance).

We now present the main result necessary for the proof of Proposition 4.

**Theorem 11** (Theorem 3.2.39 (Federer, 1996))**.** *If $S$ is a closed $m$-rectifiable subset of $\mathbb{R}^n$, then we have*

$$\mathcal{M}^m(S) = \mathcal{H}^m(S).$$

From this theorem, the following holds immediately.

**Corollary 12.** *For small $\varepsilon$, we have an asymptotic form for the Lebesgue measure of an $\varepsilon$-neighborhood of an $m$-dimensional smooth compact manifold with boundary $S$ in $\mathbb{R}^n$ as*

$$\mathrm{Vol}(\{x \mid \|x - S\| \le \varepsilon\}) = \alpha(n - m)\varepsilon^{n-m}\mathcal{H}^m(S) + o(\varepsilon^{n-m}).$$

## C.2  PROOF OF PROPOSITION 4

*Proof.* From now, we denote the $\varepsilon$-neighborhood of a set $A \subset \Theta$ as

$$A_\varepsilon := \{\theta \in \Theta \mid \|\theta - A\| \le \varepsilon\}$$

and the Lebesgue measure of $A$ on $\Theta$ as $\mathrm{Vol}(A)$. We also denote the dimension of $A$ as $\dim(A)$. We prove this theorem by showing the following two steps:

1. The sampled $\widehat{\theta}_{n,\varepsilon}$ lies in the $\varepsilon$-neighborhood of $\bar{\Theta}$ with high probability $1 - O(\varepsilon)$.

2. If $\widehat{\theta}_{n,\varepsilon}$ is sampled from this neighborhood, then the generalization error becomes no more than $(q\varepsilon)^2$.

First, we evaluate the probability of sampling $\widehat{\theta}_{n,\varepsilon}$ from the $\varepsilon$-neighborhood of $\bar{\Theta}$. Since we sample $\widehat{\theta}_{n,\varepsilon}$ from $\widehat{\Theta}_{n,\varepsilon}$ uniformly, this probability is written as

$$\mathrm{Vol}(\bar{\Theta}_\varepsilon)/\mathrm{Vol}(\widehat{\Theta}_{n,\varepsilon}).$$

We can decompose as

$$\mathrm{Vol}(\widehat{\Theta}_{n,\varepsilon}) \leq \mathrm{Vol}(\bar{\Theta}_\varepsilon) + \mathrm{Vol}((\widehat{\Theta}_n \setminus \bar{\Theta})_\varepsilon) \tag{4}$$

since $\widehat{\Theta}_{n,\varepsilon} \subset \bar{\Theta}_\varepsilon \cup (\widehat{\Theta}_n \setminus \bar{\Theta})_\varepsilon$ naturally holds. Since $\bar{\Theta}$ contains a $\dim(\bar{\Theta})$-dimensional analytic manifold, Corollary 12 shows that there exists a positive constant $c_1$ such that

$$\mathrm{Vol}(\bar{\Theta}_\varepsilon) \geq c_1 \varepsilon^{\dim(\Theta) - \dim(\bar{\Theta})} + o(\varepsilon^{\dim(\Theta) - \dim(\bar{\Theta})}). \tag{5}$$

Next, observe that $\widehat{\Theta}_n \setminus \bar{\Theta}$ is a semi-analytic set, which is a parameter set defined by equations and inequalities of analytic functions. By Section 3 in Hardt (1975), a semi-analytic set defined on a compact parameter set is decomposed into a finite number of smooth manifolds. Hence, if we decompose $\widehat{\Theta}_n \setminus \bar{\Theta}$ into smooth manifolds $\widehat{\Theta}_n \setminus \bar{\Theta} = Z^1 \cup \cdots \cup Z^k$, we have

$$\mathrm{Vol}((\widehat{\Theta}_n \setminus \bar{\Theta})_\varepsilon) \leq \sum_{i=1}^{k} \mathrm{Vol}(Z_\varepsilon^i).$$

Since each $Z^i$ has a dimension of no more than $\dim(\widehat{\Theta}_n \setminus \bar{\Theta})$, we have from Corollary 12,

$$\mathrm{Vol}((\widehat{\Theta}_n \setminus \bar{\Theta})_\varepsilon) \leq c_2 \varepsilon^{\dim(\Theta) - \dim(\widehat{\Theta}_n \setminus \bar{\Theta})} + o(\varepsilon^{\dim(\Theta) - \dim(\widehat{\Theta}_n \setminus \bar{\Theta})}) \tag{6}$$

for a positive constant $c_2$.

From the proof of Theorem 2, we have

$$\dim(\widehat{\Theta}_n \setminus \bar{\Theta}) \leq \dim(\bar{\Theta}) - 1. \tag{7}$$

Combining (4), (5), (6) and (7), we have

$$\mathrm{Vol}(\bar{\Theta}_\varepsilon)/\mathrm{Vol}(\widehat{\Theta}_{n,\varepsilon}) = 1 - O(\varepsilon).$$

From the discussion above, $\widehat{\theta}_{n,\varepsilon}$ sampled from $\widehat{\Theta}_{n,\varepsilon}$ is in $\bar{\Theta}_\varepsilon$ with probability $1 - O(\varepsilon)$.

Second, we evaluate the generalization error when we sample $\widehat{\theta}_{n,\varepsilon}$ from $\bar{\Theta}_\varepsilon$. Since $f(x; \theta)$ is $q$-Lipschitz continuous with respect to $\theta$ for every $x \in \mathcal{X}$, we have for some $\bar{\theta} \in \bar{\Theta}$,

$$\begin{aligned}
L(\widehat{\theta}_{n,\varepsilon}) &= \mathbb{E}_{x \sim \mathcal{D}} \left[ \frac{1}{2} \left\| f(x; \widehat{\theta}_{n,\varepsilon}) - f^*(x; \theta^*) \right\|^2 \right] \\
&= \mathbb{E}_{x \sim \mathcal{D}} \left[ \frac{1}{2} \left\| (f(x; \widehat{\theta}_{n,\varepsilon}) - f(x; \bar{\theta})) + (f(x; \bar{\theta}) - f^*(x; \theta^*)) \right\|^2 \right] \\
&\leq \mathbb{E}_{x \sim \mathcal{D}} \left[ \left\| f(x; \widehat{\theta}_{n,\varepsilon}) - f(x; \bar{\theta}) \right\|^2 \right] + \mathbb{E}_{x \sim \mathcal{D}} \left[ \left\| f(x; \bar{\theta}) - f^*(x; \theta^*) \right\|^2 \right] \\
&\leq (q \| \widehat{\theta}_{n,\varepsilon} - \bar{\theta} \|)^2 + 0 \\
&\leq (q\varepsilon)^2,
\end{aligned}$$

which completes the proof. $\qquad\square$

# D   Proof of Theorem 5

Obviously, the loss function of DLNN satisfies Assumption 1 because the linear transformation is analytic.

Therefore, by Theorem 2, we only have to study the lower bound of $d_{\bar{\Theta}}$. We prove by the following step:

1. Construct the subset $\bar{\Theta}'$ of TES $\bar{\Theta}$.

2. Count the number of free parameters of $\bar{\Theta}'$ and it is the lower bound of the dimension $d_{\bar{\Theta}}$ of $\bar{\Theta}$.

First, we define a subset $\bar{\Theta}'$ of $\Theta$ as follows.

**Definition 13** (Subset $\bar{\Theta}'$ of $\Theta$). We denote arbitrary matrices or vectors as $\mathcal{M}$ and arbitrary regular matrices as $\mathcal{R}$. $\bar{\Theta}'$ is the subset of $\Theta$ which satisfies the following condition. Superscripts represent the layer index and subscripts are for the purpose of distinction.

1. For the parameter of first layer,

$$w^{(1)} = \begin{pmatrix} w^{(1)^*} \\ \mathcal{M}_1^{(1)} \end{pmatrix}, b^{(1)} = \begin{pmatrix} b^{(1)^*} \\ \mathcal{M}_2^{(1)} \end{pmatrix}.$$

2. For the parameter of $\ell$-th layer ($2 \leq \ell \leq L^* - 1$),

$$w^{(\ell)} = \begin{pmatrix} w^{(\ell)^*} & \mathcal{M}_1^{(\ell)} \\ \mathcal{M}_2^{(\ell)} & \mathcal{M}_3^{(\ell)} \end{pmatrix}, b^{(\ell)} = \begin{pmatrix} b^{(\ell)^*} \\ \mathcal{M}_4^{(\ell)} \end{pmatrix}.$$

3. For the parameter of $\ell$-th layer ($L^* \leq \ell \leq L - 1$),

$$w^{(\ell)} = \begin{pmatrix} \mathcal{R}^{(\ell)} & \mathcal{M}_1^{(\ell)} \\ \mathcal{M}_2^{(\ell)} & \mathcal{M}_3^{(\ell)} \end{pmatrix}, b^{(\ell)} = \begin{pmatrix} \mathcal{M}_4^{(\ell)} \\ \mathcal{M}_5^{(\ell)} \end{pmatrix},$$

where $\mathcal{R}^{(\ell)} \in \mathbb{R}^{m_{L^*-1}^* \times m_{L^*-1}^*}$.

4. For the parameter of $L$-th layer,

$$w^{(L)} = \begin{pmatrix} w^{(L^*)^*} P^{-1} & \mathcal{M}_1^{(L)} \end{pmatrix}, b^{(L)} = b^{(L^*)^*} - q^{(L)},$$

where $P = \mathcal{R}^{(L-1)} \cdots \mathcal{R}^{(L^*)}$ and $q^{(L)}$ is a term determined by other parameters.

Now we proceed to the proof of the first step.

**Lemma 13.** $\bar{\Theta}'$ *is a subset of* $\bar{\Theta}$.

*Proof.* We only have to prove that the output of the student network whose parameter is in $\bar{\Theta}'$ is the same as that of the teacher network. We denote the output of $\ell$-th layer of the teacher network as $h^{(\ell)^*} (\in \mathbb{R}^{m_\ell^*})$ and the output in the redundant width of $\ell$-th layer as $r^{(\ell)}$.

The output of first layer of the student network is

$$\begin{pmatrix} w^{(1)^*} x \\ \mathcal{M}_1^{(1)} x \end{pmatrix} + \begin{pmatrix} b^{(1)^*} \\ \mathcal{M}_2^{(1)} \end{pmatrix} = \begin{pmatrix} w^{(1)^*} x + b^{(1)^*} \\ \mathcal{M}_1^{(1)} x + \mathcal{M}_2^{(1)} \end{pmatrix} = \begin{pmatrix} h^{(1)^*} \\ r^{(1)} \end{pmatrix}.$$

The output of second layer of the student network is

$$\begin{pmatrix} w^{(2)^*} h^{(1)^*} + \mathcal{M}_1^{(2)} r^{(1)} \\ \mathcal{M}_2^{(2)} h^{(1)^*} + \mathcal{M}_3^{(2)} r^{(1)} \end{pmatrix} + \begin{pmatrix} b^{(2)^*} \\ \mathcal{M}_4^{(2)} \end{pmatrix} = \begin{pmatrix} h^{(2)^*} + \mathcal{M}_1^{(2)} r^{(1)} \\ r^{(2)} \end{pmatrix}.$$

By continuing the same discussion, we can show that the output of $(L^* - 1)$-th layer can be written as $\begin{pmatrix} h^{(L^*-1)^*} + q^{(L^*-1)} \\ r^{(L^*-1)} \end{pmatrix}$, where we write the redundant term as $q$.

The output of $L^*$-th layer of the student network is

$$\begin{pmatrix} \mathcal{R}^{(L^*)}\left(h^{(L^*-1)^*} + q^{(L^*-1)}\right) + \mathcal{M}_1^{(L^*)} r^{(L^*-1)} \\ \mathcal{M}_2^{(L^*)}\left(h^{(L^*-1)^*} + q^{(L^*-1)}\right) + \mathcal{M}_3^{(L^*)} r^{(L^*-1)} \end{pmatrix} + \begin{pmatrix} \mathcal{M}_4^{(L^*)} \\ \mathcal{M}_5^{(L^*)} \end{pmatrix} = \begin{pmatrix} \mathcal{R}^{(L^*)} h^{(L^*-1)^*} + q^{(L^*)} \\ r^{(L^*)} \end{pmatrix}.$$

The output of $(L^* + 1)$-th layer of the student network is

$$\begin{pmatrix} \mathcal{R}^{(L^*+1)}\left(\mathcal{R}^{(L^*)} h^{(L^*-1)^*} + q^{(L^*)}\right) + \mathcal{M}_1^{(L^*+1)} r^{(L^*)} \\ \mathcal{M}_2^{(L^*+1)}\left(\mathcal{R}^{(L^*)} h^{(L^*-1)^*} + q^{(L^*)}\right) + \mathcal{M}_3^{(L^*+1)} r^{(L^*)} \end{pmatrix} + \begin{pmatrix} \mathcal{M}_4^{(L^*+1)} \\ \mathcal{M}_5^{(L^*+1)} \end{pmatrix}$$

$$= \begin{pmatrix} \mathcal{R}^{(L^*+1)}\mathcal{R}^{(L^*)} h^{(L^*-1)^*} + q^{(L^*+1)} \\ r^{(L^*+1)} \end{pmatrix}.$$

By continuing the same discussion, we can show that the output of $(L - 1)$-th layer is

$$\begin{pmatrix} \mathcal{R}^{(L-1)} \cdots \mathcal{R}^{(L^*)} h^{(L^*-1)^*} + q^{(L-1)} \\ r^{(L-1)} \end{pmatrix} = \begin{pmatrix} P h^{(L^*-1)^*} + q^{(L-1)} \\ r^{(L-1)} \end{pmatrix}.$$

Hence, the output of the last layer is

$$w^{(L^*)^*} P^{-1}(P h^{(L^*-1)^*} + q^{(L-1)}) + \mathcal{M}_1^{(L)} r^{(L-1)} + b^{(L^*)^*} - q^{(L)}$$

$$= h^{(L^*)^*} + w^{(L^*)^*} P^{-1} q^{(L-1)} + \mathcal{M}_1^{(L)} r^{(L-1)} - q^{(L)}.$$

So, if we set $q^{(L)} = w^{(L^*)^*} P^{-1} q^{(L-1)} + \mathcal{M}_1^{(L)} r^{(L-1)}$, the output is the same as that of teacher network. $\qquad\square$

We proceed to the proof of the second step.

**Lemma 14.** *The lower bound of the dimension of $\bar{\Theta}$ is $d_\Theta - d^*$.*

*Proof.* From Lemma 13, $\bar{\Theta}$ contains $\bar{\Theta}'$. The number of the free element of the parameter in $\bar{\Theta}'$ is the number of elements expressed by $\mathcal{M}$ and $\mathcal{R}$, so it is $d_\Theta - d^*$. Hence, $\bar{\Theta}$ contains a $(d_\Theta - d^*)$-dimensional hyper cube and its internal. A $(d_\Theta - d^*)$-dimensional hyper cube and its internal contain a hyper sphere and its internal, which is an analytic manifold whose dimension is $d_\Theta - d^*$. So the maximum dimension of analytic manifolds contained in $\bar{\Theta}$ is at least $d_\Theta - d^*$, which completes the proof. $\qquad\square$

# E    Proof of Theorem 6

The loss function of FCDNN satisfies Assumption 1 since the linear transformation and the activation function is analytic. We prove in the same way as Theorem 5.

**Definition 14** (Subset $\bar{\Theta}'$ of $\Theta$)**.** Let us denote arbitrary matrices or vectors as $\mathcal{M}$. $\bar{\Theta}'$ is the subset of $\Theta$ which satisfies the following condition. Superscripts represent the layer index and subscripts are for the purpose of distinction.

1. For the parameter of first layer,

$$w^{(1)} = \begin{pmatrix} w^{(1)^*} \\ \mathcal{M}_1^{(1)} \end{pmatrix}, b^{(1)} = \begin{pmatrix} b^{(1)^*} \\ \mathcal{M}_2^{(1)} \end{pmatrix}.$$

2. For the parameter of $\ell$-th layer ($2 \leq \ell \leq L - 1$),

$$w^{(\ell)} = \begin{pmatrix} w^{(\ell)^*} & 0 \\ \mathcal{M}_1^{(\ell)} & \mathcal{M}_2^{(\ell)} \end{pmatrix}, b^{(\ell)} = \begin{pmatrix} b^{(\ell)^*} \\ \mathcal{M}_3^{(\ell)} \end{pmatrix}.$$

3. For the parameter of $L$-th layer,

$$w^{(L)} = \left( w^{(L^*)^*} \quad 0 \right), b^{(L)} = b^{(L^*)^*}.$$

Now we proceed to the proof of the first step.

**Lemma 15.** $\bar{\Theta}'$ *is a subset of* $\bar{\Theta}$.

*Proof.* We only have to prove that the output of the student network whose parameter is in $\bar{\Theta}'$ is the same as that of the teacher network. We denote the **pre-activation** in $\ell$-th layer of the teacher network as $h^{(\ell)^*}$ ($\in \mathbb{R}^{m_\ell^*}$) and the **pre-activation** in the redundant width of the student network in $\ell$-th layer as $r^{(\ell)}$.

The pre-activation in the first layer of the student network is

$$\begin{pmatrix} w^{(1)^*}x \\ \mathcal{M}_1^{(1)}x \end{pmatrix} + \begin{pmatrix} b^{(1)^*} \\ \mathcal{M}_2^{(1)} \end{pmatrix} = \begin{pmatrix} w^{(1)^*}x + b^{(1)^*} \\ \mathcal{M}_1^{(1)}x + \mathcal{M}_2^{(1)} \end{pmatrix} = \begin{pmatrix} h^{(1)^*} \\ r^{(1)} \end{pmatrix}.$$

The pre-activation in the second layer of the student network is

$$\begin{pmatrix} w^{(2)^*}\sigma(h^{(1)^*}) + 0\sigma(r^{(1)}) \\ \mathcal{M}_1^{(2)}\sigma(h^{(1)^*}) + \mathcal{M}_2^{(2)}\sigma(r^{(1)}) \end{pmatrix} + \begin{pmatrix} b^{(2)^*} \\ \mathcal{M}_3^{(2)} \end{pmatrix} = \begin{pmatrix} h^{(2)^*} \\ r^{(2)} \end{pmatrix}.$$

By continuing the same discussion, we can show that the pre-activation of $(L-1)$-th layer can be written as $\begin{pmatrix} h^{(L-1)^*} \\ r^{(L-1)} \end{pmatrix}$.

Hence, the output of the last layer is

$$w^{(L)^*}\sigma(h^{(L-1)^*}) + 0\sigma(r^{(L-1)}) + b^{(L)^*} = h^{(L)^*},$$

which is the same as the output of the teacher network. $\qquad\square$

We proceed to the proof of the second step.

**Lemma 16.** *The lower bound of the dimension of* $\bar{\Theta}$ *is* $d_\Theta - \sum_{\ell=1}^{L} m_\ell^*(m_{\ell-1} + 1)$.

*Proof.* From Lemma 15, $\bar{\Theta}$ contains $\bar{\Theta}'$. The number of the free element of the parameter in $\bar{\Theta}'$ is the number of elements expressed by $\mathcal{M}$, so it is $\sum_{\ell=1}^{L}(m_\ell - m_\ell^*)(m_{\ell-1} + 1) = d_\Theta - \sum_{\ell=1}^{L} m_\ell^*(m_{\ell-1} + 1)$. Hence, $\bar{\Theta}$ contains a $(d_\Theta - \sum_{\ell=1}^{L} m_\ell^*(m_{\ell-1} + 1))$-dimensional hyper cube and its internal. A $(d_\Theta - \sum_{\ell=1}^{L} m_\ell^*(m_{\ell-1} + 1))$-dimensional hyper cube and its internal contain a hyper sphere and its internal, which is an analytic manifold of dimension $d_\Theta - \sum_{\ell=1}^{L} m_\ell^*(m_{\ell-1} + 1)$. So the maximum dimension of analytic manifolds contained in $\bar{\Theta}$ is at least $d_\Theta - \sum_{\ell=1}^{L} m_\ell^*(m_{\ell-1} + 1)$, which completes the proof. $\quad\square$

# F  DETAILS OF EXPERIMENTS

## F.1  GUESS AND CHECK ALGORITHM FOR SAMPLING RANDOM NEAR INTERPOLATORS

To obtain a predictor with a random near interpolator, we can utilize the Guess & Check (G&C) algorithm by Chiang et al. (2023), which provides a practical implementation to sample random near interpolators. Specifically, the G&C algorithm is operated by the following procedure:

1. Fix a sufficiently small $\varepsilon > 0$.
2. Sample a parameter $\theta$ according to the uniform distribution on $\Theta$, without considering the training set.
3. For the sampled parameter $\theta$, evaluate the training loss $L_n(\theta)$ using the training set $\{(x_i, y_i)\}_{i=1}^{n}$.
4. If $L_n(\theta) \leq \varepsilon$ holds, return $\theta$; otherwise, go back to 2.

**Algorithm 1:** Pattern Search Algorithm

---

**Input:** Initial point $\theta_0$, step size $\alpha_0 > 0$, stopping threshold $\varepsilon$
**Output:** Approximate solution $\theta_k$
Set $k \leftarrow 0$;
**while** $L_n(\theta_k) > \varepsilon$ **do**
    Compute trial points $\theta_k + \alpha_k \theta_{k,i(k)}$ or $\theta_k - \alpha_k \theta_{k,i(k)}$, where $i(k) \in \{1, \cdots, d_\Theta\}$ is
    randomly chosen and $\theta_{k,i(k)}$ denotes the $i(k)$-th parameter of $\theta_k$;
    **if** *there exists $i(k) \in \{1, \cdots, d_\Theta\}$ s.t. $L_n(\theta_k + \alpha_k \theta_{k,i(k)}) < L_n(\theta_k)$* **then**
        $\theta_{k+1} \leftarrow \theta_k + \alpha_k \theta_{k,i(k)}$;
        $\alpha_{k+1} \leftarrow \alpha_k$;
    **else if** *there exists $i(k) \in \{1, \cdots, d_\Theta\}$ s.t. $L_n(\theta_k - \alpha_k \theta_{k,i(k)}) < L_n(\theta_k)$* **then**
        $\theta_{k+1} \leftarrow \theta_k - \alpha_k \theta_{k,i(k)}$;
        $\alpha_{k+1} \leftarrow \alpha_k$;
    **else**
        $\theta_{k+1} \leftarrow \theta_k$;
        $\alpha_{k+1} \leftarrow \gamma_{\text{dec}}\, \alpha_k$;
    $k \leftarrow k + 1$;

---

### F.2 DETAILS OF SECTION 6.1

Below, we provide an outline of the setup. We consider a regression problem and adopt the mean squared error as the loss function. The input data $x$ is 2-dimensional vector and is generated according to the uniform distribution on $[-1, 1]^2$. The output data $y$ is a scalar output of a teacher FCDNN, which is a randomly initialized FCDNN by Xavier's uniform initialization (Glorot & Bengio, 2010) with hidden layers consisting of 5 units. The student FCDNN to be trained is a FCDNN with the same number of layers as the teacher and with hidden layers consisting of 10 units. The activation function is the hyperbolic tangent (tanh) in both the teacher and student FCDNNs. To sample random near interpolators, we run the G&C algorithm presented in the previous section $\varepsilon = 0.01$ and Xavier's uniform initialization. We then compute the test loss using 2000 samples randomly generated from the teacher FCDNN. This procedure is repeated 1000 times for each training sample size. We conduct experiments with networks consisting of 2, 4, and 6 layers, respectively.

### F.3 DETAILS OF SECTION 6.2

We adapted a cross entropy loss for the loss function. We set the learning rate of Adam as $0.001$ and other hyper-parameters as the default value of PyTorch. We estimate the dimension of $\bar{\Theta}$ by utilizing lPCA algorithm (Fukunaga & Olsen, 1971), implemented by the scikit-dimension package (Bac et al., 2021). We compute the test loss on the MNIST test split. In order to reproduce the teacher–student setting on MNIST dataset, we remove about 9 % of ambiguous data from the test split of MNIST by utilizing the cleanlab package (Northcutt et al., 2020).

### F.4 EXPERIMENT BY PATTERN SEARCH

The pattern search algorithm serves as an alternative procedure for generating random near interpolators without implicit bias, analogous to the G&C algorithm. Specifically, the algorithm perturbs the parameter in a random direction and then evaluates whether this perturbation decreases the objective function. If the update results in a decrease, the new parameter is accepted; otherwise, the parameter is reverted to its original value. Furthermore, if none of the attempted updates yield a decrease, we reduce the step size accordingly. A more detailed description of the procedure is provided in Algorithm 1.

We describe the details of our experimental setup. Because this algorithm is computationally expensive, we restrict the MNIST dataset to samples whose labels are 0 or 1. We employ a 2-layer FCDNN with a hidden-layer width of 10 and the softplus activation function. The loss function is the cross-entropy loss. The model parameters are initialized using Xavier's uniform initialization, and the step size is initialized as 1.0. We set the stopping threshold to $\varepsilon = 0.01$.

To estimate the dimension $d_{\bar{\Theta}}$ of $\bar{\Theta}$, we approximately sample from $\bar{\Theta}$ by training the 2-layer FCDNN on all MNIST samples labeled 0 or 1 using the pattern search algorithm. This procedure is repeated 10000 times, producing 10000 approximate samples from $\bar{\Theta}$. We then estimate the dimension of the manifold on which these samples lie by applying the lPCA algorithm implemented in the scikit-dimension package (Bac et al., 2021).

Next, we generate 1000 random near interpolators using the pattern search algorithm, varying the number of training samples across 1000, 2000, 3000, 4000, 5000, 6000, and 7000, and compute their test loss on the MNIST test split.

The results are presented in Figure 4. The estimated upper bound is consistent with the empirical findings, as it provides a sufficient number of samples for the generalization of random near interpolators.

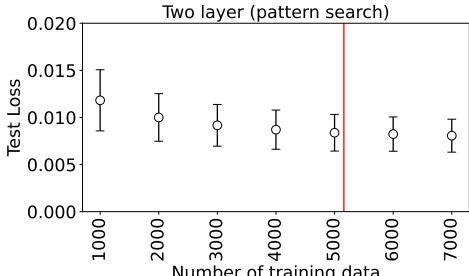

Figure 4: Test losses of random near interpolators on 2-layer FCDNN. The vertical axis represents the test loss, while the horizontal axis corresponds to the number of training data. The error bars indicate the standard deviation over 1000 trials for each training sample size. The red vertical line is the estimated upper bound of the strong sample complexity $d_{\Theta} - d_{\bar{\Theta}} + 1$.