# OpenReview forum: "Zero Generalization Error Theorem for Random Interpolators via Algebraic Geometry"
_ICLR.cc/2026/Conference — Submitted to ICLR 2026_

### Official Review · Reviewer_dTSd · 2025-10-15

**Soundness:** 2
**Presentation:** 2
**Contribution:** 1
**Rating:** 2
**Confidence:** 4

**Summary:**

This paper proves that randomly sampled interpolators (parameters that fit the training data exactly) can achieve zero generalization error once the sample size passes a finite threshold determined by the model’s geometry. The authors make two strong assumptions: (i) the data are noiseless, and (ii) the true function is representable by the network (realizability). They upper-bound the strong sample complexity by $\dim(\Theta)-\dim(\overline{\Theta})+2$, where $\Theta$ is the parameter space and $\overline{\Theta}$ is the teacher-equivalence set (TES); the claimed result holds with probability 1 under a real-analytic student model and an absolutely continuous input distribution.

**Strengths:**

The approach of upper-bounding the strong sample complexity through an algebraic-geometric lens is interesting. Similar conclusions could be formulated in function space (rather than parameter space) via Bézout-type arguments, so **the novelty appears limited**. I have a few concerns about the proofs (listed below) and would appreciate the authors’ clarifications.

**Weaknesses:**

- **Scope/novelty**: The main result (Theorem 2) closely aligns with classical dimension-counting arguments in function space: viewing the parametric model’s image inside a function space, each exact data constraint $f(x)=y$ imposes an affine hyperplane, so intersecting with $k$ such constraints reduces dimension by **at most** $k$. This suggests a finite “strong sample complexity’’ comparable to the model (image) dimension under suitable **regularity/transversality**.

- **Proposition 4**: I am not fully convinced by the tube-measure step as stated in the proof. For example, the set $\overline{\Theta}$ (TES) need not be a **smooth submanifold** and this causes a problem! This can occur, for instance, when the teacher is realizable by a highly overparameterized network, which is not uncommon. Moreover, **in Figure 1**, you correctly show TES is not a smooth manifold, which suggests that **you are aware** of the potential presence of non-smooth points. Note that Weyl’s tube formula in its classical form applies to smooth compact submanifolds (or, more generally, requires conditions like positive reach/curvature measures or a stratified argument with the top stratum dominating). Please **(a)** state assumptions ensuring smoothness/positive reach (or a Whitney stratification plus a reference that justifies the leading-order volume scaling for real-analytic sets), and **(b)** clarify how one gets equation (2) in line 768. As written, these technicalities are crucial for the $1 - O(\varepsilon)$ probability conclusion.

- **Theorem 2**: In the proof outline (lines 344–345), the reduction by $n$ is attributed to imposing $n$ equations. Are you assuming a **complete-intersection** or **transversality condition** so that each constraint generically reduces the dimension by $1$? My concern is that the data is **not generic** (since it is generated by the **true function without noise**), so the resulting equations **need not be generic**.

- **Notation**: Line 125 says “squared norm’’ but writes $\||\cdot\||$; please clarify whether you mean the Euclidean norm $\||\cdot\||_2$ or the squared Euclidean norm $\||\cdot\||_2^2$ (cf. line 150, which appears to use the squared norm).

Overall, I find this line of research interesting. However, in its current state, the work does not meet the standards of a theoretical contribution suitable for ICLR, primarily due to **incomplete (and potentially wrong) proofs**. I am not fully convinced of their correctness, and several essential details appear to be missing. This raises doubts about the validity of the results in the level of generality claimed by the authors. Nevertheless, I remain open to discussion.

**Questions:**

Can you clarify how you get equation (2) in line 768? Also can you clarify in general how one can compute the dimension of the true parameter set?

Lastly, can you explain my main concerns regarding the proofs of Theorem 2 and Proposition 4?

---

> ### Author Response · Authors · 2025-11-21
>
> We are grateful for your insightful and fruitful comments. We appreciate your acknowledgement of the interest in applying algebraic geometry to generalization–error analysis.
>
> ---
>
> >The main result (Theorem 2) closely aligns with classical dimension-counting arguments in function space.
>
> While our results may not introduce mathematically novel concepts, they incorporate arguments specific to generalization error analysis, which we believe constitute a unique aspect of our contribution.
> As pointed out, the fact that the dimension of the zero set of analytic functions decreases according to the number of equations can indeed be derived under appropriate regularity conditions by a classical argument in algebraic geometry, and is therefore not novel from a purely mathematical standpoint. However, our analysis contains a component that is unique to generalization error theory. In particular, we must show that the transversality condition holds for almost all $x_{n+1}$ in order to guarantee that the generalization error becomes zero without relying on mathematically technical assumptions. This aspect represents the distinctive and novel contribution of our analysis.
>
> ---
>
> >Proposition 4: I am not fully convinced by the tube-measure step as stated in the proof.
>
> As you correctly point out, the idea of directly applying Weyl's tube formula to analytic sets is a mathematically problematic discussion. We fix the proof by computing the lower bound of the volume of $\bar{\Theta}_\varepsilon$ by the fact that $\bar{\Theta}$ contains a $\text{dim}(\bar{\Theta})$-dimensional analytic manifold and the upper bound for the volume of ${(\hat{\Theta}_n \setminus \bar{\Theta})}_\epsilon$ by the fact that an analytic set can be decomposed into a finite number of smooth manifolds by Whitney's stratification argument. By this argument, we can fix the proof without any additional assumptions. We fix the proof in Appendix C in the revised version. We sincerely appreciate that you have carefully examined our proof and provided accurate and insightful comments.
>
> ---
>
> >Theorem 2: In the proof outline (lines 344–345), the reduction by $n$ is attributed to imposing $n$ equations. Are you assuming a complete-intersection or transversality condition so that each constraint generically reduces the dimension by $1$? My concern is that the data is not generic (since it is generated by the true function without noise), so the resulting equations need not be generic.
>
> The output data $y$ is certainly not generic, but the input data $x$ is generic, so the resulting equations become generic. In addition, we discuss the reduction of the dimension occurs almost everywhere in the parameter space, so we implicitly omit the irregular case in which the complete-intersection or transversality condition are not satisfied.
>
> ---
>
> >Notation: Line 125 says “squared norm’’ but writes $||\cdot||$; please clarify whether you mean the Euclidean norm $||\cdot||_2$ or the squared Euclidean norm $||\cdot||_2^2$ (cf. line 150, which appears to use the squared norm).
>
> As you point out, our use of the term is problematic. We mean the Euclidean norm by $||\cdot||$. We fix the term in the paper.
>
> ---
>
> >Can you clarify how you get equation (2) in line 768?
>
> As you point out, this equation does not hold. We fix it by an inequality.
>
> ---
>
> >Also can you clarify in general how one can compute the dimension of the true parameter set?
>
> Unfortunately, we do not have a general method for computing the exact dimension of the true parameter set for all the machine learning model. However, we can compute the lower bound of it in many cases by explicitly constructing the subset of true parameter set as is done in this paper for a deep linear network case (Theorem 5) or a general deep neural network case (Theorem 6). The lower bound is sufficient for applying Theorem 2 since we can compute the upper bound of the strong sample complexity.
>
> ---
>
> >Lastly, can you explain my main concerns regarding the proofs of Theorem 2 and Proposition 4?
>
> As you correctly point out, the proof of Proposition 4 was not complete, and we have fixed this issue in the revised version of the paper.
> Regarding Theorem 2, the proof is indeed correct; the generic property of the equations is ensured by the generic property of the choice of $x$, as is discussed above.

---

> ### Comment · Reviewer_dTSd · 2025-11-24
> **Concerns regarding the proofs**
>
> Thank you very much to the authors for their response.
>
> I still have some concerns regarding the proofs, and I would greatly appreciate clarification on the following points.
>
> **Concerns regarding Proposition 4**
>
> 1) In the proof of Proposition 4, you reference the original Weyl’s tube formula. As far as I know, the standard statement applies to smooth compact manifolds without boundary. Since you restrict the parameter space to a compact set, a boundary will appear. I believe this issue is fixable, but it requires invoking a correct version of the tube formula that explicitly handles manifolds with boundary. Please provide such a version, together with the full statement of the theorem. This would help substantially in understanding the argument.
>
> 2) Please clarify the constants $c_1$ and $c_2$ are positive! Referring to them simply as “constants” is not informative for the reader. It would also help to describe these constants explicitly in terms of the volume of the unit ball and the volume of the relevant geometric sets (as in Weyl’s tube formula).
>
> 3) In Part II of the proof (starting at line 797), it should be explained clearly why the factor $1-O(\epsilon)$ is needed. In addition, at the end of the proof, it is not clear how the inequalities between lines 807 and 808 follow. At the moment, the logical flow of the argument feels disconnected.
>
>
> **Concerns regarding Theorem 2**
>
> In the proof of Theorem 2, the argument relies on pulling back the data constraints to the parameter space, which is how the real analytic set is obtained. Note that, when pulling back, we also see the parametrization map.
> Now, regarding the claimed dimension reduction: once we pull back through the parametrization map, the structure of this map may prevent the dimension from decreasing as more samples are added. Even if the input data is random, nothing guarantees dimension reduction unless specific structural properties of the architecture are used. This is a major missing component of the proof. For this reason, I believe Theorem 2 should be proved for specific architectures for which these structural conditions can be verified.
>
>
> **Notation $\ell1$**
>
> Thank you for correcting the Euclidean norm. In line 125, it is written $\ell1$. Do you mean $\ell^1$ ?

---

> > ### Author Response · Authors · 2025-12-02
> >
> > >Concerns regarding Proposition 4
> >
> > **Concerns about Weyl's tube formula**
> >
> > As you correctly point out, Weyl's tube formula is not applicable to a general analytic manifold with boundary. In response to this, we have revised our proof to avoid relying on Weyl's tube formula and instead use an argument based on Minkowski's content [Federer, 1996]. This notion provides an asymptotic expression for the volume of the $\varepsilon$-neighborhood of a “general’’ manifold when $\varepsilon$ is sufficiently small.
> > Specifically, by applying Theorem 3.2.39 in [Federer, 1996], we can show that the volume of an $\varepsilon$-neighborhood of a $d'$-dimensional analytic manifold with boundary in $R^d$ can be written as
> > \begin{equation}
> > c_1\varepsilon^{d-d'}+o(\varepsilon^{d-d'})
> > \end{equation}
> > where $c_1>0$ is a constant. Using this result, we have corrected the proof accordingly in the revised version of the paper. We have also included the necessary argument for applying the Minkowski’s content approach in Appendix C.1.
> > We sincerely appreciate your careful and insightful comment.
> >
> > **Positivity of the constants**
> >
> > In the revised version of the paper, we have explicitly clarified that $c_1$ and $c_2$ are positive constants.
> >
> > **The necessity of $1-O(\varepsilon)$ evaluation**
> >
> > We need this evaluation for evaluating the probability of the event where the sampled $\hat{\theta}\_{n,\varepsilon}$ from $\hat{\Theta}\_{n,\varepsilon}$ is in the neighborhood of $\bar{\Theta}\_\varepsilon$. If $\hat{\theta}\_{n,\varepsilon}$ is sampled from this neighborhood, the generalization error becomes small since it is near $\bar{\Theta}$, where the generalization error becomes $0$. We have added this logical flow at the beginning of the proof to improve clarity.
> >
> > ---
> >
> > >Concerns regarding Theorem 2
> >
> > We understand that your concern is that our choice of $x_{n+1}$ in the proof depends on $\hat{\Theta}\_n$, and therefore depends on the previously chosen samples $x_1, ..., x_n$. Consequently, the sequence $x_1,...,x_{n+1}$ constructed in this manner might not be generic, or more precisely, might not be generated as i.i.d. samples from $\mathcal{D}$. From this perspective, your observation is indeed correct, and we acknowledge that the previous version of the proof was insufficient. We show that even when $x_{n+1}, ..., x_1$ are generated i.i.d., the probability that the dimension of $\hat{\Theta}\_n$ becomes $d_\Theta-n$ is equal to 1. The core of the proof is the fact that, for any choice of $x_1,\ldots, x_n$, one can always select an $x_{n+1}$ with probability 1 that reduces the dimension of $\hat{\Theta}\_n$ by one. Using this fact, we can apply the argument based on conditional probabilities  inductively, which establishes the desired result. Since the proof is a bit long, please see the revised version of the paper for more detail (step2 of the proof in Appendix B contains the relevant discussion).
> >
> > ---
> >
> > >Notation $\ell1$
> >
> > We fixed the notation.

---

### Official Review · Reviewer_Codj · 2025-10-24

**Soundness:** 3
**Presentation:** 4
**Contribution:** 3
**Rating:** 6
**Confidence:** 3

**Summary:**

Under a teacher–student regression setting, the authors prove that the generalization error of randomly sampled interpolators becomes exactly zero once the number of training examples exceeds a threshold determined by the geometry of the interpolator set in parameter space. Using tools from algebraic geometry—specifically real analytic sets—the paper introduces and rigorously analyzes the concept of strong sample complexity.

**Strengths:**

- **Clear and Well-Written Presentation:** The paper is generally well-written, and the results are clean.
- **Innovative Theoretical Contribution:** The paper rigorously analyzes the generalization properties of interpolators using tools from algebraic geometry. The derived results are both elegant and insightful. This paper is solid.
- **Empirical Validation:** Experimental results on synthetic regression tasks and the MNIST dataset back up the theoretical findings, showing that the predicted bounds on the strong sample complexity are consistent with observed generalization behavior.

**Weaknesses:**

- **Strong & Restrictive Assumptions:** The analysis is carried out in a controlled, noiseless teacher–student setting, and the student model is assumed to be real analytic. In practical scenarios, these assumptions may not hold, e.g., the teacher model is a ReLU MLP.
- **Limited Applicability to General Machine Learning Settings:** The theoretical results depend on the assumption that the teacher and student models belong to the same parametric function class—more precisely, Assumption 2 requires the teacher-equivalent set (TES) to be non-empty. In many real-world machine learning tasks, the 'teacher model' (not in the teacher-student model) typically has a larger or more expressive function class compared to the student model. Consequently, Assumption 2 may fail to hold, thereby limiting the applicability of the results to more general settings.

Overall, I think this paper has a significant contribution to the teacher-student setting. It deserves to be published!

**Questions:**

In general machine learning settings, the generalization error is typically of the order $O(1 / \sqrt{n})$, where $n$ is the sample size. Consequently, to achieve a generalization error smaller than a given threshold $\epsilon$, the required $n$ often depends exponentially on the input dimension $d$. In contrast, the sample complexity to achieve zero generalization error presented in this paper is much lower. While we acknowledge that this work is situated within a teacher-student framework, it would significantly enhance the contribution of the paper if the authors could further discuss the following points:
- (1) Can the results derived in this paper be extended to a more general machine learning setting beyond the teacher–student model?
- (2) What is the relationship between the proposed sample complexity results and the traditional generalization bounds (e.g., the $O(1 / \sqrt{n})$ rate) encountered in broader machine learning contexts?
- (3) Is it possible to extend these results to non-analytic models, such as neural networks with ReLU activations? If not, what are the main obstacles that prevent such an extension?

A detailed discussion addressing these questions would not only clarify the scope of the current results but also broaden the impact of the work.

---

> ### Author Response · Authors · 2025-11-21
>
> Thank you for fruitful and insightful comments. We appreciate your acknowledgement of our use of algebraic geometry in generalization–error analysis, and we are grateful for your recognition that this approach leads to elegant and insightful theorems.
>
> ---
>
> >The analysis is carried out in a controlled, noiseless teacher–student setting, and the student model is assumed to be real analytic.
>
> The noiseless setting can potentially be relaxed. Let $f$ denote the function learned from noiseless data and $\hat{f}$ the function learned from data containing $O(\epsilon)$ noise. In this case, the two learned functions satisfy
> \begin{equation}
> ||f-\hat{f}||\le O(\epsilon).
> \end{equation}
> According to our analysis, the generalization error of $f$ becomes zero under certain conditions. Consequently, the generalization error of $\hat{f}$ is bounded by $O(\epsilon)$, indicating that our framework naturally extends to settings with small amounts of noise.
>
> ---
>
> >The theoretical results depend on the assumption that the teacher and student models belong to the same parametric function class—more precisely, Assumption 2 requires the teacher-equivalent set (TES) to be non-empty.
>
> Assumption 2 does not require that the teacher and student models belong to the same parametric function class; rather, it assumes that the function class represented by the student model contains that of the teacher model. Although the reviewer suggests that the teacher model may, in some cases, belong to a larger function class than the student, we believe that this is not the typical situation in modern machine learning. In current practice, student models are generally overparameterized, and their expressive capacity is strong enough to represent functions from a wide variety of function classes, as suggested by universal approximation results.
>
> Furthermore, it is widely believed that many real-world datasets possess an underlying low-dimensional structure, implying that the corresponding teacher functions are comparatively simple. This intuition is reflected in Theorems 5 and 6, where the teacher network is a smaller neural network and the student is substantially larger. For these reasons, we consider it reasonable to assume that the function class expressed by the student model contains that of the teacher model, even in practical machine learning settings.
>
> ---
>
> >Can the results derived in this paper be extended to a more general machine learning setting beyond the teacher–student model?
>
> From a discussion above, we can relax the teacher-student setting to a noisy situation.
>
> ---
>
> >What is the relationship between the proposed sample complexity results and the traditional generalization bounds (e.g., the $O(1/\sqrt{n})$ rate) encountered in broader machine learning contexts?
>
> It is difficult to directly compare our results with those from traditional generalization analyses. Most traditional generalization bounds do not analyze parameters that achieve exactly zero training error. Instead, they typically focus on parameters with small norms or on parameters obtained through specific optimization algorithms. In contrast, our analysis concentrates on parameters that exactly interpolate the data, and it leads to the conclusion that a finite number of samples can yield zero generalization error.
>
> That said, there are some conceptual connections. For example, the discussion regarding the relaxation of the noiseless setting suggests a potential link to classical bounds. In addition, our evaluation of the dimensionality of the hypothesis set is closely related to covering number arguments commonly used in traditional generalization theory.
>
> ---
>
> >Is it possible to extend these results to non-analytic models, such as neural networks with ReLU activations?
>
> Unfortunately, we were not able to extend our result to non-analytic functions such as ReLU. The key difficulty is that the behavior of zero sets differs fundamentally between analytic and non-analytic cases. We illustrate this issue with a simple example.
> Consider a student model given by a 1-width, two-layer neural network with a ReLU activation,
> \begin{equation}
> f(x;\theta)=w_2\text{ReLU}(w_1x+b_1), w_1,w_2,b_1\in R
> \end{equation}
> and the teacher model
> \begin{equation}
> f^*(x)=\text{ReLU}(x), x\in[-1,1].
> \end{equation}
> When $x>0$, the equation $w_2\text{ReLU}(w_1x+b_1)=x$ must hold, yielding the constraints $w_2w_1=1$ and $b_2=0$ and $w_1>0$ in the true parameter set. Thus, the true parameter set lies on a one-dimensional manifold.
> However, if the sampled data $x_1, x_2, ...,x_n$ all lie in [-1,0] (this probability is positive since $x$ is generated uniformly), then it suffices to satisfy $w_2=0$. In this case, the dimension of $\hat{\Theta}_n$ remains to be $2$, regardless of how large the dataset becomes. This pathological behavior arises because ReLU contains a flat (constant) region, making it unnecessary for the parameter set to contract further.

---

### Official Review · Reviewer_LkHH · 2025-10-27

**Soundness:** 3
**Presentation:** 2
**Contribution:** 3
**Rating:** 8
**Confidence:** 3

**Summary:**

This paper proposes a new theory to explain the good generalization error of neural networks in the interpolation regime and in the teacher-framework setting. More specifically, The authors assume that the data (both training and test) is generated by a teacher network, without noise. Then, the generalization error of student network is studied, by focusing on the weight vectors achieving zero training error (assuming these always exist). The paper is based on the analysis of the dimension of these interpolators set, by assuming they form real analytics manifold. Based on this analysis, the authors managed to get a bound on the sample complexity, effectively showing that if the number of data points is large enough, then he empirical interpolator set converges to the ``population'' interpolator set, and, hence, all interpolators of the training errors achieve zero test error (for $n$ large enough). The technique is also extended to near interpolators. Finally, the theory is supported by numerical experiments.

**Strengths:**

- The authors propose a model-based analysis of the generalization error of interpolators in a teacher-student setting. This offers an interesting perspective on generalization, showing that perfect generalisation error can be reacher when there is structure in the data distribution and the model is compatible with it (in the sense that it can interpolate).
- The paper uses tools from algebraic geometry, suggesting new links between this field and statistical learning theory
- The literature review in the introduction seems quite complete

**Weaknesses:**

*Main weaknesses:*
 - The introduction put an emphasis on the models that "employ an excessive number of parameters". However, the proposed theory states that the strong sample complexity is bounded by the number parameters of the student network. This bound seems to suggest that not too much overparameterization is allowed in order to obtain zero generalization error. Even in Theorem 6, the derived sample complexity seems to be $k = O(\sqrt{d_\Theta})$, which allows some but not arbitrary overparameterization. While the bounds are very interesting, I think that the link with overparameterization should be discussed more.
 - Line 401, it is stated that the strong sample complexity is independent on the number of parameters, but it still depends on the with of the student network, this should be clarified.
 - In my opinion, the claim of the abstract to explain the generalization error of interpolators "for general machine learning models" is a bit strong, given that the study is restricted to the teacher-student setting, which imposes a lot of structure on the learning task that is not necessarily present in general. I think this point should be clarified.
 - The assumption of real analyticity might prevent some modern neural architectures to fall into this framework.


*Other issues:*
- Line 190, the uniform distribution on $\widehat{\Theta}_n$ is mentioned, but it is not clear to me in general that a set can be endowed with a uniform distribution. What is meant by uniform distribution should maybe be clarified.

**Questions:**

- A lot of recent work also suggest that is an implicit bias of the learning algorithm towards well-generalizing solutions, can you discuss how compatible these findings are with your theory?
- Do you think that your theory can be extended to ReLU networks, where the analyticity assumptions might fail?
- Why is there a +2 in theorem 2? In the proof sketch, it seems to be a +1 (of course it is a very minor change)?

---

> ### Author Response · Authors · 2025-11-21
>
> Thank you for a insightful and valuable comments. We appreciate your positive feedback on our demonstration of the interesting fact that perfect generalization can be achieved when the data possess sufficient structure.
>
> ---
>
> >The introduction put an emphasis on the models that "employ an excessive number of parameters". However, the proposed theory states that the strong sample complexity is bounded by the number parameters of the student network.
>
> As the reviewer correctly points out, our current results do not fully address all overparameterized scenarios. To avoid potential misunderstanding, we have revised the expression “overparameterized model” to “large model” in the paper. Nevertheless, we would like to emphasize that Theorem 5 does capture an overparameterized setting: the required sample complexity depends only on the number of parameters in the teacher network and is independent of the number of parameters in the student network. This demonstrates that our analysis can accommodate certain forms of overparameterization.
>
> ---
>
> >Line 401, it is stated that the strong sample complexity is independent on the number of parameters, but it still depends on the with of the student network, this should be clarified.
>
> As you point out, the bound in Theorem 6 depends on the number of parameters. We delete this statement.
>
> ---
>
> >In my opinion, the claim of the abstract to explain the generalization error of interpolators "for general machine learning models" is a bit strong, given that the study is restricted to the teacher-student setting, which imposes a lot of structure on the learning task that is not necessarily present in general.
>
> As you point out, the word "general" is over-claimed and confusing. We revise the sentence.
>
> ---
>
> >Line 190, the uniform distribution on $\hat{\Theta}_n$ is mentioned, but it is not clear to me in general that a set can be endowed with a uniform distribution. What is meant by uniform distribution should maybe be clarified.
>
> We consider the parameter set $\Theta$ to be compact. Hence $\hat{\Theta}_n$ is also compact and we can consider a uniform distribution on it.
>
> ---
>
> >A lot of recent work also suggest that is an implicit bias of the learning algorithm towards well-generalizing solutions, can you discuss how compatible these findings are with your theory?
>
> We believe that the line of research on implicit bias and our work are largely independent. Our paper focuses on the universal behavior exhibited by all interpolators and does not address the bias that may arise from selecting a particular interpolator. As noted earlier, this viewpoint is supported by empirical findings such as [Chiang et al, ICLR 2023], which demonstrate that randomly sampled interpolators tend to generalize well.
>
> If we were to highlight a possible connection, one natural example is the case in which a learning algorithm possesses a stationary distribution, such as gradient Langevin dynamics. Sampling from such a stationary distribution effectively corresponds to the random sampling of interpolators considered in our work. In this setting, the implicit bias of the learning algorithm can be interpreted through the lens of our theoretical results.
>
> ---
>
> >Do you think that your theory can be extended to ReLU networks, where the analyticity assumptions might fail?
>
> Unfortunately, we were not able to extend our result to non-analytic functions such as ReLU. The key difficulty is that the behavior of zero sets differs fundamentally between analytic and non-analytic cases. We illustrate this issue with a simple example.
> Consider a student model given by a 1-width, two-layer neural network with a ReLU activation,
> \begin{equation}
> f(x;\theta)=w_2\text{ReLU}(w_1x+b_1), w_1,w_2,b_1\in R
> \end{equation}
> and the teacher model
> \begin{equation}
> f^*(x)=\text{ReLU}(x), x\in [-1,1].
> \end{equation}
> When $x>0$, the equation $w_2\text{ReLU}(w_1x+b_1)=x$ must hold, yielding the constraints $w_2w_1=1$ and $b_2=0$ and $w_1>0$ in the true parameter set. Thus, the true parameter set lies on a one-dimensional manifold.
> However, if the sampled data $x_1, x_2, ...,x_n$ all lie in $[-1,0]$ (this probability is positive since $x$ is generated uniformly), then it suffices to satisfy $w_2=0$. In this case, the dimension of $\hat{\Theta}_n$ remains to be $2$, regardless of how large the number of the data becomes. This pathological behavior arises because ReLU contains a flat (constant) region, making it unnecessary for the parameter set to contract further.
>
> ---
>
> >Why is there a +2 in theorem 2? In the proof sketch, it seems to be a +1 (of course it is a very minor change)?
>
> This point arises from a very minor technical issue. Our analysis applies dimensionality arguments to an analytic set through a sequence of reductions; however, the original set $\Theta$ is not itself analytic. For this reason, we begin the reduction from $\hat{\Theta}1$, which we have to assume to be $d_\Theta$-dimensional.

---

> > ### Author Response · Authors · 2025-12-02
> >
> > >Why is there a +2 in theorem 2? In the proof sketch, it seems to be a +1 (of course it is a very minor change)?
> >
> > We have improved the bound from $a+2$ to $a+1$ by incorporating an additional argument into the proof in the revised version of the paper.

---

### Official Review · Reviewer_p2LG · 2025-10-29

**Soundness:** 2
**Presentation:** 2
**Contribution:** 1
**Rating:** 2
**Confidence:** 4

**Summary:**

The paper studies the generalization performance of a random interpolator. Under certain technical conditions, the main of which being a sufficient number of samples, the authors prove that the latter generalization error equals exactly 0.  The authors use it to make a point that the implicit bias property, studied a lot in the recent literature, might not be crucial for explaining generalization for the over-parameterized models. The paper uses techniques from algebraic geometry in the proofs of its main results.

**Strengths:**

The problem formulation addresses generalization - one of the cornerstones of modern machine learning. The approach is novel and interesting.

**Weaknesses:**

The resulting bounds appear to be vacuous for the overparameterized models, even though analyzing the latter serves as the motivation in the introductions. To corroborate further, consider the simplest case where both the teacher and the student networks match each other and are given by $f(w, x) = wTx$, where $x, w \in \mathbb{R}^d$. Denote the frozen weight parameter of the teacher network by $w_*$. Take $x$ to be i.i.d. standard normal. Then, first of all, the only zero generalization error weight is $w = w_*$.  As such, the theorem can be true only if it's the only interpolator. This is indeed the case: if $n \ge d + 2$, there are enough equations in the corresponding system to recover $w_*$ uniquely. **However,  $n \ge d + 2$ corresponds to an under-parametrized scenario**, missing the point of the minimum l_2-interpolator framework.  And I suspect that the same would happen for the non-linear neural networks. To add more to it, it is known that, for highly over-parameterized settings, the generalization error of an interpolator can depend heavily on the choice of the interpolator, as demonstrated extensively, for example, in "Stochastic Mirror Descent on Overparameterized Nonlinear Models: Convergence, Implicit Regularization, and Generalization" by Azizan et al.

**Questions:**

1. Do you have any examples where Theorem 2 is applicable to an over-parameterized problem?
2. How do we sample an interpolator if the set of all interpolators is not compact, such as in the case of a linear student and a linear teacher? I think this is not addressed in the paper, as it just says that the parameter is sampled "uniformly".
3. What exactly does experiment in Section 6.2 validate if we still use Adam, which likely has its own implicit bias, to find all interpolators we consider? Also, it is very easy to attain a near-zero accuracy on MNIST-10, but what would the experiment look like for more complicated datasets, such as cifar-10?

---

> ### Author Response · Authors · 2025-11-21
>
> Thank you for the valuable and insightful comment. We appreciate your positive feedback regarding the novelty of our contribution to generalization–error analysis.
>
> ---
>
> >The resulting bounds appear to be vacuous for the overparameterized models, even though analyzing the latter serves as the motivation in the introductions.
>
> The vacuity pointed out only appears in certain important but limited models, such as linear models and convex losses. As the reviewer correctly points out, when the model $f$ is linear, Theorem 2 only holds under the condition $n \ge d + 2$, which indeed results in a vacuous bound. However, this situation corresponds to a very simple setting known as a non-singular model, where $\bar{\Theta}$ reduces to a singleton ${w_*}$, and this case is already addressed in Corollary 3.
> In contrast, most modern machine learning models, including neural networks and Transformers, are singular models, in which $\bar{\Theta}$ is not a singleton but instead forms a high-dimensional manifold. For example, Theorem 5 analyzes deep linear neural networks in which the student model is over-parameterized relative to the teacher network. It shows that the bound requires only $n \ge d^* + 2$, where $d^*$ denotes the number of parameters in the teacher network. This result naturally accommodates over-parameterized scenarios, as the required number of data points depends solely on the size of the teacher network, independent of the size of the trained (student) network.
> Furthermore, Theorem 6 addresses models with general activation functions and establishes a requirement of $n \ge O(\sqrt{d})$, which also effectively handles over-parameterized regimes.
> We appreciate this important point. To clarify this point, we have added a remark below Theorem 2 in section 4.1, explaining its relationship between our general result with Corollary 3, which deals with situations such as linear models.
>
> ---
>
> >To add more to it, it is known that, for highly over-parameterized settings, the generalization error of an interpolator can depend heavily on the choice of the interpolator, as demonstrated extensively, for example, in "Stochastic Mirror Descent on Overparameterized Nonlinear Models: Convergence, Implicit Regularization, and Generalization" by Azizan et al.
>
> We agree with the reviewer that, as pointed out, the generalization properties of an interpolator can indeed depend on the specific choice of interpolator, and it is certainly possible to construct interpolators that do not generalize well. However, the focus of this paper is on the universal behavior shared by all interpolators, rather than on the bias introduced by any particular choice of an interpolator. This perspective is supported by empirical findings such as those reported in [Chiang et al, ICLR 2023], which demonstrate that randomly sampled interpolators tend to generalize effectively.
>
> ---
>
> >Do you have any examples where Theorem 2 is applicable to an over-parameterized problem?
>
> Theorem 5 for a deep linear neural network case, and Theorem 6 for a general deep neural network case are good examples.
>
> ---
>
> >How do we sample an interpolator if the set of all interpolators is not compact, such as in the case of a linear student and a linear teacher? I think this is not addressed in the paper, as it just says that the parameter is sampled "uniformly".
>
> In this paper, we restrict the parameter space to be compact and do not address the non-compact case. Therefore, when the set of all interpolators is itself non-compact, we sample interpolators from the subset that lies within the compact parameter space. That said, in practical experiments, it is rare to encounter cases in which training is only possible if the parameters diverge to infinity. Therefore, considering the theory within a compact space is therefore also meaningful from a practical standpoint.
>
> ---
>
> >What exactly does experiment in Section 6.2 validate if we still use Adam, which likely has its own implicit bias, to find all interpolators we consider?
>
> We adopted Adam because sampling truly random interpolators over a high-dimensional parameter space is computationally expensive. Nevertheless, during the rebuttal period, we are attempting to sample interpolators more faithfully at random by using a pattern search algorithm on the 2-labeled MNIST dataset.
>
> ---
>
> >Also, it is very easy to attain a near-zero accuracy on MNIST-10, but what would the experiment look like for more complicated datasets, such as cifar-10?
>
> We chose to use MNIST due to computational limitations. Specifically, estimating the dimension of the true parameter set requires sampling a number of interpolators comparable to the number of parameters in the model, which makes it impractical to complete such an experiment within the rebuttal period. Nevertheless, as this is a theoretical paper, we believe that an experiment on MNIST is sufficient to support our theoretical results.

---

> > ### Author Response · Authors · 2025-12-02
> >
> > >What exactly does experiment in Section 6.2 validate if we still use Adam, which likely has its own implicit bias, to find all interpolators we consider?
> >
> > We added an additional experiment to Appendix F.4 using the pattern search algorithm, which is an implicit-bias-free method that performs random search in the parameter space. Nevertheless, in this experiment, the outcomes produced by the random search algorithm and those produced by gradient-based methods such as Adam are highly similar, suggesting that the implicit bias induced by SGD plays only a minimal role. Therefore, employing Adam to sample random near interpolators is justified in the context of this experiment. We have added this justification for the use of Adam for sampling random near interpolators in Section 6.2.

---

### Author Response · Authors · 2025-12-02

Dear Reviewers,

We sincerely appreciate your thoughtful and insightful feedback. In response to your comments, we have revised the manuscript as summarized below. All revised or newly added sentences are highlighted in blue in the updated version.

---

**The applicability of our result to over-parametrized cases**

- We have added an explanation below Theorem 2 discussing its applicability to over-parameterized settings, where the number of model parameters exceeds the number of data samples.

**Fixed proof**

- We have corrected the incomplete proofs of Theorem 2 and Proposition 4 in Appendices B and C, respectively.
- Related to this, we have summarized in Appendix C.1 the prerequisite background needed to prove Proposition 4.

**Improved bound**

- We have strengthened the bound on the strong sample complexity in Theorem 2, improving it from $d_\Theta-d_{\bar{\Theta}}+2$ to $d_\Theta-d_{\bar{\Theta}}+1$ , by adding the necessary argument to Appendix B.

**Additional experiment**

- We have added an experiment using the pattern search algorithm to avoid the implicit bias induced by SGD, conducted under conditions similar to those in Section 6.2.
- We have added an explanation in Section 6.2 justifying the use of Adam, which is an algorithm that inherently possesses implicit bias, for sampling random interpolators.

**Other minor revisions**

- We have corrected several expressions, including the use of the term “over-parametrized,” and refined some notations for clarity.

---

### Author Response · Authors · 2025-12-03

Dear Area Chair(s),

Since the discussion phase concluded before we were able to fully address the remaining points, we would like to summarize the key issues that persisted after our partial exchanges with the Reviewers. Two reviewers provided positive evaluations of our paper, while the remaining two raised mainly following concerns in their reviews:

- **Non-applicability to over-parameterized situations** (Reviewer p2LG): The reviewer noted that our result appears applicable only to under-parameterized settings, as the bound becomes vacuous in the linear model case. While it is true that our bound is vacuous for regular models such as linear models, it becomes non-vacuous for statistically singular models, including deep neural networks. This is demonstrated in Theorems 5 and 6. The validity of our bound for singular models is one of the central contributions of our work, and we would like to emphasize that our results are indeed **applicable to over-parameterized situations**. Further clarification can be found in our response to Reviewer p2LG.

- **Incomplete proofs** (Reviewer dTSd): The reviewer constructively remarked that they “remain open to discussion,” specifically regarding the incomplete proofs of Theorem 2 and Proposition 4. Upon review, we found that although the results themselves were correct, the original proofs lacked sufficient detail. We have revised and completed the proofs in the Appendix. Additional explanation is provided in our final response to Reviewer dTSd.

We hope that this summary helps clarify the remaining points of concern, and we appreciate your consideration.

---

### Meta-Review · Area_Chair_fV1k · 2026-01-05

**Summary:**

This paper studies the generalization behavior of a random interpolator and proves, under some technical conditions (i.e., a sufficiently large sample size), that the generalization error can be exactly zero.
Using tools from algebraic geometry, the authors argue that implicit bias may not be essential for explaining generalization.

All reviewers agree that the core idea is interesting, making this a difficult decision.
However, the current analysis does not adequately address truly overparameterized regimes, does not extend to practical non-analytic models such as ReLU networks, and still suffers from limitations in both proof rigor and presentation.
As a result, the paper requires substantial revision before it can make a sufficiently strong and convincing contribution, and it cannot be accepted in its current form.

**Reviewer Concerns:**

- Reviewer **p2LG** asked for: (1) clarification of the theoretical results in the overparameterized regime (which is of clear greater interest),  and (2) stronger empirical results in appropriate settings. I believe these points were addressed only partially during the rebuttal.
- Reviewer **LkHH** asked for: (1) more discussion and/or clarification of the results for overparameterized models, (2) a more appropriate presentation of the results and contributions, and (3) whether the results extend beyond real and analytic functions (e.g., to ReLU networks). I believe that points (1) and (3) remain outstanding after the rebuttal.
- Reviewer **Codj** asked for: (1) further extension of the results beyond the teacher–student setup, and (2) extensions to practical non-analytic models. I believe that point (1) was partially addressed, while point (2) remains outstanding.
- Reviewer **dTSd** identified technical issues in the proofs of the main results and iterated with the authors to resolve some of them.

**Reviewer Scores:**

- I believe that reviewer **p2LG** would have slightly increased their score or kept it unchanged (e.g., 2 -> 4), since some of the concerns were partially addressed during the rebuttal.
- I believe that reviewer **LkHH** would have kept their score unchanged (8, which is already highly positive).
- I believe that reviewer **Codj** would have kept their score unchanged (6, which is already positive).
- I believe that reviewer **dTSd** would have slightly increased their score or kept it unchanged (e.g., 2 -> 4), as not all concerns were fully addressed.

---

### Decision · Program_Chairs · 2026-01-26

Reject